# Potential Applications of NRF2 Modulators in Cancer Therapy

**DOI:** 10.3390/antiox9030193

**Published:** 2020-02-25

**Authors:** Emiliano Panieri, Aleksandra Buha, Pelin Telkoparan-Akillilar, Dilek Cevik, Demetrios Kouretas, Aristidis Veskoukis, Zoi Skaperda, Aristidis Tsatsakis, David Wallace, Sibel Suzen, Luciano Saso

**Affiliations:** 1Department of Physiology and Pharmacology “Vittorio Erspamer”, Sapienza University of Rome, 00185 Rome, Italy; luciano.saso@uniroma1.it; 2Department of Toxicology “Akademik Danilo Soldatović”, University of Belgrade-Faculty of Pharmacy, 11000 Belgrade, Serbia; aleksandra.buha@pharmacy.bg.ac.rs; 3Department of Medical Biology, Faculty of Medicine, Yuksek Ihtisas University, 06520 Balgat, Ankara, Turkey; pelinta@yiu.edu.tr (P.T.-A.); dilekcevik@yiu.edu.tr (D.C.); 4Department of Biochemistry-Biotechnology University of Thessaly Viopolis, Mezourlo, 41500 Larissa, Greece; dkouret@uth.gr (D.K.); veskoukis@uth.gr (A.V.); skaperda@uth.gr (Z.S.); 5Laboratory of Toxicology Science and Research, Medical School, University of Crete, 71003 Heraklion, Crete, Greece; tsatsaka@uoc.gr; 6School of Biomedical Science, Oklahoma State University Center for Health Sciences, Tulsa, OK 74107-1898, USA; david.wallace@okstate.edu; 7Department of Pharmaceutical Chemistry, Faculty of Pharmacy, Ankara University, 06100 Tandogan, Ankara, Turkey; Sibel.Suzen@pharmacy.ankara.edu.tr

**Keywords:** NRF2-KEAP1, ROS, cancer metabolism, antioxidant, oxidative stress, cancer therapy, chemoresistance, radioresistance

## Abstract

The nuclear factor erythroid 2-related factor 2 (NRF2)–Kelch-like ECH-associated protein 1 (KEAP1) regulatory pathway plays an essential role in protecting cells and tissues from oxidative, electrophilic, and xenobiotic stress. By controlling the transactivation of over 500 cytoprotective genes, the NRF2 transcription factor has been implicated in the physiopathology of several human diseases, including cancer. In this respect, accumulating evidence indicates that NRF2 can act as a double-edged sword, being able to mediate tumor suppressive or pro-oncogenic functions, depending on the specific biological context of its activation. Thus, a better understanding of the mechanisms that control NRF2 functions and the most appropriate context of its activation is a prerequisite for the development of effective therapeutic strategies based on NRF2 modulation. In line of principle, the controlled activation of NRF2 might reduce the risk of cancer initiation and development in normal cells by scavenging reactive-oxygen species (ROS) and by preventing genomic instability through decreased DNA damage. In contrast however, already transformed cells with constitutive or prolonged activation of NRF2 signaling might represent a major clinical hurdle and exhibit an aggressive phenotype characterized by therapy resistance and unfavorable prognosis, requiring the use of NRF2 inhibitors. In this review, we will focus on the dual roles of the NRF2-KEAP1 pathway in cancer promotion and inhibition, describing the mechanisms of its activation and potential therapeutic strategies based on the use of context-specific modulation of NRF2.

## 1. Introduction

Nuclear factor erythroid 2-related factor 2 (NRF2) is a key transcription factor and a key modulator of cellular antioxidant responses that regulates the expression of genes encoding antioxidant enzymes with a protective role against different types of oxidative changes. NRF2 in combination with its own negative regulator, Kelch-like ECH-associated protein 1 (KEAP1), has become the center of a debate regarding whether NRF2 suppresses the tumor promotion or, conversely, exerts pro-oncogenic functions. Based on this, the present review will describe the role of NRF2 in cancer prevention and promotion, discussing potential advantages and disadvantages derived from its therapeutic modulation in cancer prevention and treatment. As it is known, under non-stressed conditions, NRF2 is constitutively poly-ubiquitinated by the CUL3-KEAP1 E3 ubiquitin ligase complex and subjected to degradation through the proteasome pathway [1]. After exposure to several redox altering stimuli, highly reactive thiols of KEAP1 are subjected to instant modification, leading to NRF2 stabilization caused by its decreased affinity for the CUL3-KEAP1 complex. Subsequently, NRF2 translocates into the nucleus and binds to the antioxidant response element (ARE) located within the promoter region of specific target genes, inducing the expression of a large number of cytoprotective proteins with antioxidant and detoxifying roles [2,3,4,5]. The importance of NRF2 function has been demonstrated by several studies using NRF2-deficient mice showing an increased susceptibility to redox disturbances and xenobiotic stress [6,7,8]. It has also been shown that tissue oxidative damage after ischemia and reperfusion is efficiently counteracted by NRF2 induction [9]. In line with the protective roles of the KEAP1-NRF2 pathway, its activation seems to effectively prevent carcinogenesis by promoting a number of antioxidant mechanisms [10,11]. Thus, NRF2 activation may have beneficial role as a result of its suppressive effect on carcinogenesis. On the other hand, increasing evidence show that constitutive NRF2 activation contributes to the progression of various cancer types. Specifically, many studies have shown that the increased activation of NRF2 in cancer cells leads to its augmented transcriptional activity and promotes tumor progression [12,13,14], metastasis formation [15], resistance to chemo-radiotherapy [16,17,18,19], and is clinically associated with poor prognosis [20]. During the last decade, several mechanisms through which NRF2 signaling pathway is persistently activated in different types of cancers have been discussed. Regarding NRF2, its oncogenic activity promotes cancer cell growth and proliferation, suppression of cancer cell apoptosis, self-renewal of cancer stem cells, therapy resistance, increased angiogenesis and anti-inflammatory activities [21]. As the pro-tumorigenic role of this factor has gained interest, pharmacological modulation of the NRF2 pathway offers pioneering therapeutic opportunities against several diseases. Recent studies have brought to light a few small molecules displaying promising properties in NRF2 inhibition, but their applicability still needs to be further investigated [22,23,24,25]. Most of these molecules lack specificity and have off-target toxic effects since they easily react with cysteine residues of different molecules. Metabolic instability, low bioavailability, and poor membrane permeability are some of the basic drawbacks in the administration of many NRF2 inhibitors [26]. To date, dimethyl fumarate is the only NRF2 activator approved by the Food and Drug Administration (FDA) but its function in cancer prevention has not been examined yet.

### Structure and Function of NRF2 and KEAP1

Nuclear factor erythroid 2-related factor 2 (NRF2) is a transcriptional factor encoded by the *NFE2L2* gene that belongs to “Cap’N’Collar” type of basic region leucine zipper factor family (CNC-bZIP) [27]. Human NRF2 protein is 605 amino acids long and contains seven conserved NRF2-ECH homology domains known as Neh1-Neh7 [27,28]. Neh2 is a major regulatory domain located to N-terminus of NRF2 and it has two binding sites known as DLG and ETGE. These sites help to regulate NRF2 stability by interacting with the Kelch domains of E3 ubiquitin ligase Kelch-like ECH-associated protein 1 (KEAP1), a substrate of Cullin 3-based ubiquitin E3 ligase complex that ubiquitinates and targets NRF2 for proteasomal degradation [29,30,31,32]. The Neh1 and Neh6 domains have also been shown to control NRF2 stability. The Neh1 contains a basic leucine zipper motif that is also known as DNA binding domain and it enhances NRF2 transcriptional activation [27,33]. The Neh6 domain is a serine-rich domain containing two motifs (DSGIS and DSAPGS) that negatively modulate NRF2 stability through beta-TrCP dependent but KEAP1 independent regulation [34]. The Neh3, Neh4, and Neh5 domains are known as trans-activation domains of NRF2. The carboxy-terminal Neh3 domain binds to CHD6 (a chromo-ATPase/helicase DNA-binding protein) that is the transcriptional co-activator of NRF2 [35]. The Neh4 and Neh5 domains interact with the CH3 domains of CBP (CREB-binding protein) that facilitates transactivation of NRF2 target genes [36,37]. In addition, a seventh domain of NRF2 known as Neh7 has been shown to interact with a nuclear receptor retinoic X receptor alpha (RXRa) that inhibits NRF2 target genes transcription [38]. A schematic representation of NRF2 structure is shown in Figure 1A.

KEAP1 is a 69-kDa protein, which belongs to the BTB-Kelch family of proteins [39]. All the members of this family assemble with Cullin-RING ligases that catalyze general protein ubiquitylation [39]. KEAP1 contains five domains including N-terminal region, the Cullin3 binding broad complex, tramtrack and broad complex/tramtrack/bric-a-brac (BTB) homodimerisation domain, the intervening region (IVR), the Kelch/double glycine repeat (DGR) domain and C-terminal domain [40,41]. The BTB domain is critical for KEAP1 dimerization and CUL3 assembly requires a BTB protein motif for ubiquitination and proteasomal degradation of NRF2 [41]. In addition, the BTB domain also contains a critical cysteine residue (Cys151) that has an important role in the activation of NRF2 [42]. The IVR/BACK domain contains highly reactive cysteines, namely Cys273 and Cys288 that function as a sensor for NRF2 inducers and are essential for controlling NRF2 activity [43]. Kelch/DGR domain negatively regulates NRF2 activation by interacting with conserved carboxyl terminus of Neh2 domain [44]. NRF2 controls antioxidant and cytoprotective genes expression and regulates cellular defense [23]. Under physiological conditions, NRF2 is kept at low levels in normal tissues by KEAP1-dependent ubiquitination and proteasomal degradation [29,30]. NRF2 localizes in the cytosol where ETGE and DLG motifs on its Neh2 domain associate with KEAP1 Kelch domain [45]. A schematic representation of KEAP1 structure is shown in Figure 1B. KEAP1 acts as an adaptor for NRF2 binding to the KEAP1-CUL3-E3 ubiquitin ligase complex, an event followed by rapid NRF2 proteasomal degradation [28]. Under oxidative stress (OS) conditions or in the presence of other stressors including reactive-oxygen species (ROS) or electrophiles, the activity of KEAP1 is decreased by direct modification of reactive cysteine residues in the IVR domain. These redox changes induced on KEAP1 thiols alter its structure and prevent KEAP1-mediated NRF2 ubiquitination [46,47]. Subsequently, NRF2 accumulates in the nucleus where it induces the expression of its target genes. In the nucleus, NRF2 heterodimerizes with small Maf proteins through Neh1 domain and promotes the antioxidant responsive elements (AREs)-dependent expression of cytoprotective genes [46] (see Figure 2). In this regard, over 500 NRF2 target genes have been so far identified and this number is expected to increase in the next coming future [48,49]. The genetic products of NRF2 activation can be categorized in functional categories covering multiple cellular processes including phase I, II, III drug/xenobiotic metabolism and detoxification, antioxidant response and redox homeostasis regulation, iron homeostasis, cellular metabolism, DNA repair, transcriptional activation, cell proliferation, and apoptosis [46,50]. A selected list of NRF2 target genes is presented in Table 1. 

## 2. NRF2 in Cancer Prevention and Therapeutic Implications

### 2.1. Therapeutic Modulation of NRF2-KEAP1 Pathway for Cancer Prevention

Historically, the NRF2-KEAP1 pathway has been the focus of extensive research aimed at assessing its potential role in human chronic diseases characterized by alterations of the redox homeostasis such as diabetes, cardiovascular diseases, neurodegenerative diseases and cancer. Among them, effort have been made to explore the chemopreventive properties of naturally occurring as well as synthetic compounds functioning as NRF2 activators or KEAP1 inhibitors in vitro and in vivo. The identification of the molecular mechanisms underlying NRF2 modulation has driven a renovated interest in the field of basic and clinical cancer research, fostering a growing number of studies. However, it is now increasingly recognized that NRF2 can exert oncogenic as well as oncosuppressive functions, so that the development of effective therapeutic approaches based on NRF2 modulation requires a careful evaluation of the specific context of its activation including not only the histotype, stage, and genetic background of a specific tumor but also the therapeutic scheme of administration and the target population that might benefit from treatment. In the following section, we will describe some of the most relevant NRF2 activators and their use in cancer treatment. 

### 2.2. Activators of NRF2

Activation of the NRF2 system is complex and can follow canonical and non-canonical pathways. A difficulty in identifying activators and inhibitors of NRF2 or KEAP1 as modulators of inflammation and potential protectors against oxidative stress and carcinogenesis, is the dual nature of the NRF2-KEAP1 protein-protein interaction [26,59,60,61,62]. Generally, activation of NRF2 has been viewed as therapeutic, but recent evidence has suggested that this event can be pro-oncogenic as well, depending on the context of NRF2 activation [62]. For example, an increased NRF2 activity under “normal” conditions will lead to improved cell defense against carcinogenesis. By contrast, unrestrained NRF2 activation in a tumorigenic condition, can be protective against stressful conditions and promote therapy resistance [62]. In cases where NRF2 activation exerts pro-tumoral effects, the therapeutic applicability of NRF2 inhibitors has been explored. Since the NRF2-KEAP1 pathway can sense electrophiles as potential stressors, the use of electrophilic drugs to induce its activation would be a reasonable starting point [41,63]. There is however a substantial concern over potential side effects associated with the use of electrophilic compounds. This concern has stimulated the development of other “modulators” of NRF2 activity, considering that the optimal compound would not be a strong NRF2 activator since the strength of its activation is proportional to the electrophilic effects [64]. More work is needed to determine the potential of NRF2 activators as therapeutic drugs, but first, the pathways involved must be better elucidated [65].

#### 2.2.1. Electrophilic and Non-Electrophilic NRF2 Activators

Collectively, NRF2-inducers fall into two classes—electrophilic and non-electrophilic [66]—with the majority of the currently identified inducers belonging to the former class. The mechanism of action for electrophilic NRF2 activators involves the interaction with the cysteine residues on KEAP1, resulting in a conformational change that releases NRF2 to its active conformational state. In the following subsections we will describe both types of molecules, but emphasizing that electrophilic activators are inherently toxic and when used at sufficiently high doses, will cause electrophilic cellular damage beyond NRF2 activation [66]. Despite the high risk for side effects, the quest for additional electrophilic covalent NRF2 activators remains [67] since the advantage of these molecules is due in part to the extremely high binding energy elicited by covalent binding compared to non-covalent and the discovery of a low-molecular weight compound able to maintain a high binding affinity is more likely to occur with covalent activators.

#### Electrophilic/Covalent NRF2 Activators

The largest class of chemicals so far identified with the ability to activate NRF2 through KEAP1 inhibition is represented by triterpenoids [68]. These molecules can bind to KEAP1 and induce a conformational change that prevents its association with NRF2, promoting its target genes transactivation. Among the others, 2-cyano-3,12-dioxooleana-1,9(11)-diene-28-oic acid (CDDO) is a synthetic derivative of the natural triterpenoid oleanolic acid with a very potent (low nanomolar doses) activating capacity on various NRF2-regulated proteins [66]. CDDO has received much attention due to its ability to hamper the development of certain tumors [66,69]. Despite KEAP1 contains 15 cysteine residues susceptible of modification by electrophilic compounds, each electrophile targets its unique series of residues, a phenomenon referred to as the “cysteine code” [70]. A key cysteine in the binding of triterpenoids to KEAP1 is C151 [70,71,72]. However, in order to improve potency, specificity and reduce potential side effects of CDDO, Methyl ester (CDDO-Me) and imidazole (CDDO-Im) derivatives have been subsequently developed. Both compounds have shown promising results, as they are able to activate NRF2 and stimulate ARE-expression at low doses [73,74]. Of note, the ability of the triterpenoids electrophilic region to react with thiol groups of many proteins and other thiol-containing molecules underscores the potential side effects associated with their use. For example CCDDO-Im has been shown to bind to mitochondrial glutathione (GSH), resulting in GSH depletion, increased oxidative stress, and increased NRF2 activation [75]. Although triterpenoids have therapeutic action at relatively low concentrations, the risk of serious side effects cannot be excluded. 

The compound D3T (3*H*-1,2-dithiole-3-thione) has been shown to increase the nuclear accumulation of NRF2, an effect mediated in part by the activation of the ERK 1/2 pathway [76]. ERK 1/2 inhibition blocked the activation of NRF2 and the effects observed on other ARE-induced gene expression were similar for oltipraz, another dithiolethione, as well as the natural NRF2-activator, sulforaphane [76]. NRF2 also controls the expression of several isoforms of the multidrug resistance-associated protein (MDR), a molecular pump that extrudes chemicals outside the cells. Indeed, it has been show that the administration of oltipraz or butylated hydroxyanisole resulted in a clear upregulation in the MDRs expression as a consequence of NRF2 activation [77]. 

#### Non-Electrophilic/Non-Covalent NRF2 Activators

Recent studies have examined the therapeutic benefits provided by non-covalently bound non-electrophiles as NRF2 activators, since electrophilic compounds can have side effects and reduce the activity of certain proteins [78]. Binding studies revealed that cysteine 151 (C151) was an important target since non-covalent binding to this site promoted cell protection, while in contrast covalent binding to this residue enhanced cell toxicity [78]. Zhang et al., recently undertook a comprehensive analysis of nearly 200 chemicals to isolate potential non-electrophilic activators of NRF2. The list was initially shortened to 86 candidates and subsequently further reduced to only 22 molecules, after the exclusion of compounds having an electrophilic reactive site. Based on their structures, the chemicals were placed into one of seven groups consisting of: I—phenothiazine; II—tricyclics; III—trihexyphenidyl; IV—phenyl pyridine; V—quinolin-8-substituted; VI—tamoxifen substituted; and VII—hexetidine. The results indicated that each class was able to induce some level of NRF2-mediated increase in ARE-dependent protein expression [79]. These changes varied between class and protein, but the systematic approach used to identify biologically active non-electrophilic NRF2 activators opened many prospects for future development. Another non-covalent small molecule activator of NRF2, RA839, was shown to activate several pathways related to NRF2 signaling in bone marrow macrophages [80]. Investigations aimed to identify, develop, and then marketing a new non-covalent NRF2 activator have been hindered by the low affinity and low potency of existing compounds compared to the covalent agents. However, one potential activator with low toxicity, but still therapeutic utility is monomethylfumerate (MMF) [59]. Another fumarate-based compound that has demonstrated promise is dimethyl fumarate (DMF) [81,82]. In addition to screening strategies of existing chemicals based on their structure and their potential use as non-electrophilic NRF2 activators, other studies are underway to find novel compounds. Chemicals that have a naphthalene moiety have shown promise as non-electrophilic NRF2 activators. Exposure of lung epithelial cells to various naphthalene-based chemicals led to a marked increase in the expression of several antioxidant enzymes, such as quinone oxidoreductase (NQO1) and heme oxygenase 1 (HO-1), elicited by NRF2 activation [83]. Few of these new compounds were of similar potency to sulforaphane, an electrophilic NRF2 activator, in stimulating the expression of these antioxidant proteins.

#### 2.2.2. Natural Compounds

A 2016 review systematically examined the activation of NRF2 by dietary factors, but also extended these findings to discuss how diet changes may restrict the nutritional utility of some activators [84]. A natural marine-based compound, honaucin A, obtained from cyanobacteria, has been reported to have anti-inflammatory properties both in vivo and in vitro. Only recently have investigators begun to focus on the pathways activated by this compound. Honaucin A forms a covalent bond with the sulfhydryl groups on KEAP1, resulting in the activation of NRF2 [85]. Many food-based NRF2 activators have been shown to work in multiple organs such as liver [86], lung [87], kidney [88], brain [89] and gastrointestinal tract [90]. Most of these NRF2 activating compounds are phenol, polyphenol, or triterpenoid. The use of compounds found in ordinary foods and spices has been an area of great interest in the last 20–25 years. For example, the spice curcumin, from turmeric and ginger family, has been reportedly used to treat a variety of disorders. Yet, little work has been done to fully understand the pathways associated with the therapeutic benefits induced by these chemicals. Recently, an entire book was dedicated to the beneficial actions of curcumin (*The Molecular Targets and Therapeutic uses of Curcumin in Health and Disease*, 2007). Several investigators focused on the molecular targets of curcumin, including NRF2, on curcumin action, and its uses as a neuroprotective agent against toxicants inducing oxidative stress but also as an antitumor agent [91,92,93]. The molecular action is non-specific with multiple systems and pathways being affected, including the involvement of NRF2 [94]. Other compounds have been examined with mixed results including silibinin, the active chemical from the milk thistle, and resveratrol, a biologically active compound from the skins of various grapes and berries. Both of these molecules have been limited in their usefulness in humans, due to mostly equivocal findings. Resveratrol has been reported to attenuate oxidative damage in the liver by increasing the expression and activity of antioxidant enzymes [95,96]. Silibinin has been shown to decrease metastasis by decreasing the activation of the PI3K-Akt and MAPK (mitogen-activated protein kinase) pathways in the lung [97], and to enhance apoptosis in colon cancer [98]. From previous reports, the effects of resveratrol and silibinin on the NRF2 pathway appear to be debatable and might depend on direct as well as indirect effects.

Foods containing compounds with positive effects on human health are sometimes referred to as “nutraceuticals.” One chemical class found in cruciferous vegetables is represented by the isothiocyanates, that include compounds such as sulforaphane, isolated from broccoli, cauliflower, cabbage, and Brussel sprouts. The role of sulforaphane in ameliorating various health-related disorders has received a lot of attention, and some of its biological effects were described in 1992 [99]. The utility of a given compound in a clinical setting is dictated by several properties such as its bioavailability, potency, and interactions with other drugs. Some aspects of the sulforaphane effects within the body suggest that there could be some level of clinical utility under certain circumstances [100]. A recent review describes the multiple pathways affected by sulforaphane administration in reducing tumor growth, indicating that the NRF2-KEAP1 pathway was a critical targeted [101]. Sulforaphane is highly electrophilic due to a reactive carbon in the isothiocyanate group that readily reacts with many nucleophiles containing a sulfur, nitrogen, or oxygen center. By targeting the sulfhydryl groups of KEAP1, sulforaphane can non-covalently bind to these reactive groups, resulting in NRF2 activation [101]. Sulforaphane can also activate antioxidant response elements (AREs) associated with NRF2 [102]. Some of the activated redox regulators include glutathione S-transferase, catalase, glutathione peroxidase, and peroxiredoxins. To improve the functionality of food-based chemicals, it is common to use the parent compound as the “backbone” and then to develop modified versions of the parent endowed with greater efficacy, potency, and possibly also reduced side effects. An example is a modified sulforaphane, 6-methylsulfinylhexyl isothiocyanate (6-HITC), which was more potent than sulforaphane in increasing the expression and activity of various AREs in lung epithelial cells [103]. Curiously, not all the effects of sulforaphane are mediated by the NRF2-KEAP1-ARE pathway. Indeed, sulforaphane can inhibit multiple inflammasomes, sensor systems that activate pro-inflammatory mediators, as it was recently shown in macrophages and fibroblasts [104]. The mechanisms of action for sulforaphane and isothiocyanate is complex and may involve multiple mediators. A critical consideration is also the cell/organ type taken into consideration. Additional research is needed to help to clarify some of the nutraceutical pathways to better assess their in vivo benefits.

### 2.3. Potential Use of NRF2 Activators in Cancer Therapy

Whereas new findings indicate that NRF2 plays a dual role in cancer ([26,105,106], the potential use of NRF2 activators in cancer prevention and therapy needs to be further elucidated. It is widely accepted that effective chemoprevention should encompass induction of cytoprotective and detoxifying enzymes. Therefore, the use of compounds able to activate NRF2-KEAP1 pathway and induce genes involved in antioxidant defense appears to be a possible strategy in both cancer prevention and therapy [64]. As explained above, NRF2 activators fall into the class of electrophilic and non-electrophilic compounds [66] and can be of natural origin or semisynthetic/synthetic analogs [64]. As reviewed by Sanders et al. [107], phenolic and sulfur-containing compounds are the most promising agents in cancer prevention. Phenolic compounds such as curcumin and resveratrol exert their chemopreventive effect via activation of NRF2-KEAP1 signaling that induces phase-II detoxifying and antioxidant enzymes (20638930). Curcumin, a common spice obtained from the rhizomes of *Curcuma longa* (turmeric), was shown to induce the expression of antioxidant enzymes such as glutathione S-transferase, aldose reductase, and HO-1 through NRF2-KEAP1 signaling [108]. Apart from covalent modification of KEAP1 [109], activation of upstream kinases such as MAPK seems to be an additional mechanism of NRF2 activation [110,111]. However, the therapeutic success of this compound has been hampered by its limited bioavailability and rapid metabolism, the poor pharmacokinetic properties [112] and the lack of conclusive toxicity data [113]. Some studies reporting therapeutic uses of curcumin in various diseases including cancer are illustrated in Table 2. Resveratrol is a natural compound contained in edible plants and fruits such as grapes, peanuts, berries, and soy with the reported capacity to increase the NRF2 levels and promote its nuclear translocation. Resveratrol monomer has been shown to induce phase-II detoxifying enzymes by activating NRF2 signaling in several human cancer cell lines [114,115] and to protect from carcinogenicity derived from bioactivated carcinogens [116]. In contrast, the effects induced by its dimers are poorly understood although the monomer and the dimers have been reported to act differently in terms of NRF2/ARE induction [117]. Table 2 illustrates some recent in vitro and in vivo studies reporting the chemotherapeutic use of resveratrol. However, similarly to curcumin, its poor bioavailability and rapid clearance, made it necessary to develop analogs with improved pharmacokinetic properties and higher potency [64]. On the other hand, green tea polyphenols such as (-)-epigallocatechin-3-gallate (EGCG) and (-)-epicatechin-3-gallate (ECG) are known NRF2 activators showing potent induction of ARE-mediated luciferase activity [118]. EGCG potentiates cellular defense capacity against chemical carcinogens, UV, and oxidative stress via NRF2-mediated induction of genes codying for antioxidant or phase-II detoxifying enzymes, modulators of inflammation, cell growth, apoptosis, cell adhesion etc. [119]. However, it has been shown that EGCG has dual effects on NRF2-mediated ARE activation depending on its concentration, with higher doses producing down-regulation and lower doses enhancing HO-1 expression [118,120]. Table 2 summarizes the results of some studies exploring the potential role of EGCG in the treatment of different pathological conditions. Other potential chemopreventive agents that exert their properties through NRF2/ARE pathway are sulfur-containing compounds, such as sulforaphane, contained in cruciferous vegetables and diallyl sulfide derived from garlic [121,122]. In comparison with curcumin and resveratrol, sulforaphane exhibits more potent activation of NRF2 and significantly better bioavailability due to its lipophilic nature and low molecular weight [123,124,125]. Preclinical and clinical evaluation of sulforaphane in breast chemoprevention revealed the presence of its metabolites in the rat mammary gland after a single oral administration at concentrations known to alter gene expression and also in human breast tissue after a single dose of broccoli sprout in healthy women undergoing reduction mammoplasty [126]. These findings provided a strong rationale for evaluating the protective effects of a broccoli sprout preparation, claiming sulforaphane as a good candidate in the adjuvant therapy based on natural molecules against several types of cancer [127]. However, some studies indicate that this compound might exert pro-survival effects in cancer cells [128] and potentially interfere with the successful application of immunotherapy [129]. Table 2 contains some of the studies conducted in this field. In a recent review [130] discussing potential combinations of a conventional anticancer drug (cisplatin or doxorubicin) and a known antioxidant (sulforaphane or curcumin), it has emerged the necessity of preclinical evidence confirming that the natural compounds can potentiate the anti-cancer effect of traditional drugs but also reduce the side toxicity in normal tissues. A review by Robledinos-Anton et al. [65] provides the information on current clinical trials in progress based on NRF2 activators and their potential clinical use in various chronic disorders, including cancer. Namely, sulforaphane is in the phase-II clinical trial for the use in treatment of prostate cancer [131] (NCT01228084) and for breast cancer [132] (NCT00843167), while it has entered the phase-II for lung cancer prevention (NCT 03232138). Also, curcumin has entered phase-III for the treatment of prostate cancer [133] (NCT02064673) and resveratrol is in the phase-I for the treatment of colon cancer [134] (NCT00256334). In an attempt to improve their biological activity, many semisynthetic and synthetic NRF2 activators have been synthesized and these substances are preferentially used in clinical practice compared to the natural counterpart. For example, Bardoxolone methyl (CDDO-Me), a semisynthetic triterpenoid with the ability to protect cells and tissues from oxidative stress by increasing the NRF2 transcriptional activity, has demonstrated its efficacy as an anticancer drug in different mouse model [135]. So far, three clinical trials focusing on the use of CDDO-Me in cancer treatment have been registered. In a phase-I clinical trial investigating the tolerability, safety, efficacy and pharmacokinetics of CDDO-Me in advanced solid tumors and lymphoid malignancies, complete response was observed in a patient with mantle cell lymphoma and partial response in a patient with anaplastic thyroid carcinoma [136]. However, phase-III of the clinical trial designed to investigate the efficacy of CDDO-Me in patients with stage 4 chronic kidney disease and type 2 diabetes revealed significant increase of heart failure within four weeks of treatment [137]. In any case, since no evidence of direct cardiotoxicity was found [138,139], trials on this compound are still ongoing. Another triterpenoid analog, omaveloxolone, has been selected for cancer treatment in three clinical trials, out of which one is ongoing. Early clinical trials for the treatment of melanoma and NSCLC have been giving promising results [140]. Dimethyl fumarate (DMF) is a synthetic NRF2 activator which has already been used for the treatment of multiple sclerosis [141] and is currently tested in a number of clinical trials, mainly investigating its efficacy in lymphoma, leukemia and melanoma. However, it should be emphasized that DMF can also exert NRF2-independent effects, suggesting that its activity might also rely on alternative pathways as evidenced by a recent study [142]. Finally, Oltipraz is an organosulfur compound which has also entered clinical trials, namely phase-I trial to study its efficacy in preventing lung cancer in smokers (NCT00006457). However, further clinical trials are needed to confirm or challenge its possible use as a chemopreventive agent. In conclusion, while the list of natural, semisynthetic and synthetic NRF2 activators is steadily increasing, it is evident that the drug development is moving slowly due to the pleiotropic effects of NRF2 activators. More studies on detailed molecular mechanisms are necessary for their possible application in cancer chemoprevention, especially in consideration of the possible oncogenic role of NRF2 in cancer cells. 

## 3. NRF2 in Cancer Promotion and Therapeutic Implications

### 3.1. Pro-Oncogenic Roles of the NRF2-KEAP1 Pathway

Given that NRF2 promotes cell survival in stress conditions [143], it is consequently accepted that enhanced NRF2 activity can be tumor promoting through several molecular mechanisms that protect cancer cells (Figure 3). This is a way by which cancer cells gain advantages over the normal cells, such as enhanced tumorigenic capacity, resistance to therapeutic agents and increased antioxidant activity leading to “NRF2 addiction” that turns this cellular guardian into a cancer driver [3]. Several important studies suggest that common oncogenes, such as *KRAS*, *BRAF*, and *MYC*, can directly promote the transcription of NRF2 through the modulation of signaling pathways such as the Raf-MEK-ERK-Jun cascade [14,53,144]. This overactivation of NRF2 leads to enhanced cytoprotective activity and, remarkably, to a decrease in ROS levels. Thus, probably oncogenes partly boost cancer development through an NRF2-dependent creation of a more favorable intracellular milieu for tumor cells selection. Constitutive activation of NRF2 in cancer promotion and the mechanisms that lead to this condition, are under debate. Many researchers have spotted cancer-associated mutations that activate NRF2 [145,146,147,148,149,150]. Mutations in NRF2 that lead to gain-of-function can be detected mainly in squamous cell carcinomas of the oesophagus, lung, larynx, and skin [151]. Additionally, aberrant NRF2 activation in cancer cells leads to remarkably increased expression of TKT and G6PD metabolic enzymes that contribute to metabolic reprogramming and cell proliferation [53]. Other evidence indicates that deficiencies in autophagy, and therefore activation of NRF2 and overexpression of p62, might promote induction of hepatic tumors [152]. As for the hormone related cancers, it has been reported that specific hormones lead to significant upregulation of NRF2 in ovarian cancer cell lines [153]. Epigenetic modifications seem to control the expression of KEAP1/NRF2 system and therefore it is important to investigate this type of regulation in cancer [154,155,156]. Another major topic is the enhancement of chemoresistance and radioresistance in cancer cells. For example, overactivation of NRF2 by pretreatment with a synthetic antioxidant was found to increase the survival of neuroblastoma cells treated with three chemotherapeutic drugs [157]. Finally, it has been demonstrated that radiation therapy leads to generation of ROS and depletion of GSH, frequently causing enhanced synthesis of antioxidant enzymes such as GCLC, HO-1 and TXRD1 by NRF2 activation, as reported in a recent study on prostate cancer cells [158]. One question that needs to be addressed, however, is whether the increase in NRF2 levels is a key step in cancer development. The existing body of evidence suggests that *KRAS* and *BRAF* increase the levels of JUN that in turn binds to well-known transcription starting sites of NRF2 promoting its induction. This finding suggests that some of these effects can be direct [14]. Moreover, many findings indicate that ROS levels should be suppressed in order to prevent cancer development due to their involvement in promoting and sustaining carcinogenesis [62,159]. It has also been proposed that utilizing drugs that boost ROS production can be an effective way of killing cancer cells [160]. Concomitantly, it is of utmost importance to determine the specific pathways and the equilibrium between ROS and NRF2 so as to elucidate the paradoxical role of KEAP1/NRF2 pathway in cancer. A schematic illustration of the pro-oncogenic functions of NRF2 is presented in Figure 3. In the following sections we will describe more in detail some of the most relevant hallmarks of cancer cells that are regulated by NRF2 activation.

#### 3.1.1. Sustained Proliferation

The KEAP1-NRF2 pathway is a vital defense system against oxidative and electrophilic stress in normal cells as well as cancer cells that use it to foster their unrestricted growth [3]. Temporary activation of NRF2 is critical for the survival of non-cancerous cells and protection against carcinogenesis [161]. However, constant activation of this pathway is detrimental especially in a cancerous context since NRF2 exerts a pro-tumoral function by supporting sustained cancer cells proliferation through various mechanisms [3]. Studies with lung cancer, pancreatic cancer, and hepatocellular carcinoma cell lines showed that NRF2-KEAP1 status directly affects their proliferation rates since NRF2-negative cells proliferate slower, while KEAP1-deleted cells proliferate faster than their wild type counterparts [162,163,164]. Moreover, NRF2 was seen to promote oncogenic *K-RasG12D*-initiated tumor formation and proliferation of pancreatic and lung cancers in vivo and *K-RasG12D*, *BRAF V619E*, and *MYC*-induced NRF2 transcription [14,165]. The role of NRF2 on cancer cell proliferation relies on the functions of the genes regulated by its own transcriptional activation [166]. The genes that regulate the proliferative capacity of normal cells such as *NOTCH1*, *NPNT*, *BMPR1A*, *IGF1*, *ITGB2*, *PDGFC*, *VEGFC*, and *JAG1* are known NRF2 targets and contribute to cancer cells survival [51]. On top of these genes, NRF2 also regulates the expression of genes needed to fulfill the constant demand of protein synthesis of cancer cells such as *PHGDH*, *PSAT1*, *PSPH*, *SHMT1*, and *SHMT2* by activating the critical effector ATF4 [51,167]. Apart from increased protein synthesis, highly proliferating cells also require energy and small building blocks to synthesize other macromolecules [168]. In this context, NRF2 regulates the expression of enzymes such as *G6PD*,*TKT*, *TALDO1*, *PPAT*, *MTHFD2*, *IDH1*, and *ME1* in lung cancer cells [53]. Furthermore, NRF2 is involved in the regulation of genes required for the synthesis of GSH (reduced glutathione) and NADPH (reduced Nicotinamide adenine dinucleotide phosphate), two crucial molecules for cell proliferation [169]. Besides glucose metabolism, NRF2 also regulates genes involved in fatty acid and lipid metabolism [170]. There is also evidence of indirect involvement of NRF2 on cancer cell proliferation by regulating several non- coding microRNAs, such as mir-1 and miR-206. These miRNAs normally inhibit *TKDT* and *G6PD* genes, and their repression by HDAC4 through NRF2 supports cancer cells growth [171]. In summary, it is clear that the increased activation of NRF2 allows cancer cells to proliferate faster as a consequence of cytoprotective genes induction and metabolic reprogramming [168]. Due to the advantages granted by its activation, the cancer cells acquire a phenotype of “NRF2 addiction” which is characterized by aberrant NRF2 accumulation in both murine and human cancers [3]. Thus, the impairment of NRF2 pathway is expected to repress tumor growth, and this is the basis of developing drugs against NRF2 in a context-dependent manner for the targeted therapy of various cancers [172]. 

#### 3.1.2. Angiogenesis Induction

The presence of constantly growing cells in tumor microenvironment causes depletion of oxygen and nutrients and creates an urgent need for a continuous supply of blood flow to fulfill the increased metabolic demand and to remove wastes and carbon dioxide [173]. Hypoxic tumor microenvironment induces the expression of *VEGF* through the transcription factor HIF-1a for the generation of new blood vessels, a process known as angiogenesis [174]. Other than VEGF, PDGF, angiopoietin, angiogenin and extracellular matrix elements participate also to the regulation of angiogenesis [175]. NRF2 modulates angiogenesis by itself and through its targeted genes [176]. Depletion of NRF2 decreases the levels of HIF-1a, which in turn causes reduction of blood vessels formation through regulation of VEGF in rat gastric epithelial cells, glioblastoma and colon cancer xenograft models [177,178,179]. In recent studies, it was reported that regulation of VEGF by NRF2 also depends on PI3K/AKT/mTOR pathways in endothelial cells [180,181]. These data suggest that both proliferative and pro-angiogenic effects of PI3K/AKT/mTOR and NRF2 pathways cooperate in favor of cancer cells. NRF2 also indirectly regulates HIF-1a levels by preventing its proteasomal degradation by the aid of NQO1 [161,182]. Angiogenesis is also regulated by common effectors of NRF2 and HIF1a including HO-1, platelet-derived growth factor (PDGFC), and fibroblast growth factor (FGF2) [183,184,185]. Among these genes, *HMOX1* was shown to promote angiogenesis in various cancers such as glioma, pancreatic cancer, and melanoma [176,186]. Intriguingly, there is an interplay between NRF2 and HIF1-a pathways since VEGF can in turn activate NRF2 via ERK1/2 signaling [187]. Another evidence supporting the crosstalk between NRF2 and HIF-1a pathway comes from the recent data showing that PIM kinases, upstream regulators of HIF-1a, promote NRF2 nuclear accumulation in response to hypoxia and in normoxia, leading to increased cancer cells survival in the hypoxic tumor microenvironment [188]. In conclusion, the induction of angiogenesis can be counted as one of the critical roles of NRF2 in promoting tumorigenesis. 

#### 3.1.3. Resistance to Apoptosis

NRF2 protects healthy cells from endogenous ROS and is a critical regulator of drug metabolism and antioxidant enzymes [21,46]. NRF2 leads to diminished apoptosis and increased drug resistance [189,190]. Inhibition of NRF2 signaling enhances apoptosis in response to oxidative insults [105,191]. Conversely, activation of NRF2 by chemopreventive agents decreases the number of apoptotic cells [192,193,194]. There are many studies showing elevated expression of NRF2 in various types of tumors such as non-small cells lung cancer (NSCLC), esophageal squamous cell cancer (ESCC), gastric cancer (GC), head and neck cancer (HNC), breast cancer (BC), ovarian cancer (OC), and endometrial cancer (EC) [46]. NRF2 signaling is activated during malignant transformation in response to radiotherapy/chemotherapy and it protects cancer cells from cell death by upregulating a number of ROS-scavenging enzymes that counterbalance the increased ROS production [195]. Moreover, NRF2 allows the cancer cells to escape death by cooperating with other pathways playing a role in apoptosis regulation. For instance, the tumor suppressor p53 inhibits NRF2 signaling by down regulating the expression of NRF2 target genes such as *x-CT*, *NQO1*, and *GST1* and triggers cell cycle arrest and apoptosis [196,197,198]. Under mild cellular stress conditions, the p21 protein, a major p53 target, binds to the DLG motif and prevents KEAP1-mediated NRF2 proteasomal degradation, activating the antioxidant response [199]. Additionally, other studies have shown that mutant p53 leads to constitutive NRF2 activation without affecting its expression and enhances cancer cells survival [200,201]. Glutathione-S-transferase pi 1 (GSTP1) is one of the downstream targets of NRF2 that inhibits proapoptotic c-Jun N-terminal kinases (JNKs) activity and promotes cell survival [202]. Importantly, p62 is another NRF2 target, which mediates autophagic degradation of KEAP1 and therefore enhances NRF2 stability, suppressing apoptosis [203]. Finally, Bcl-2 is a well-known anti-apoptotic protein that promotes increased cell survival and drug resistance [204,205]. NRF2 upregulates *Bcl-2* expression through direct binding of the ARE sequence on its promoter, which induces oncogenesis [206]. All of these studies indicate that NRF2 plays a critical role in tumor survival and drug resistance through the inhibition of apoptosis via different pathways.

#### 3.1.4. NRF2 Signaling in Metastasis

Epithelial to mesenchymal transition (EMT) is a biological process that contributes to cancer metastasis and tissue invasion [207,208]. During EMT, the expression of E-cadherin and other epithelial phenotype-related genes is repressed while conversely the expression of mesenchymal phenotype-related genes such as vimentin and N-cadherin is activated by EMT regulators (Snail, Slug, Twist, Zeb, etc.) [209,210]. As a result, epithelial cells turn into mesenchymal cells by losing their cell–cell adhesion and cell polarity features and acquiring invasive and metastatic properties. Constitutively active NRF2 has been shown in human cancers with higher metastatic capacity [211]. In addition, the correlation of NRF2 expression with cancer progression, metastasis and drug resistance has been reported in many different studies [14,15,46,212]. NRF2 promotes EMT and invasion in pancreatic adenosquamous carcinoma cells through downregulation of E-cadherin gene expression [213]. NRF2 knockdown (KD) increases E-cadherin expression and downregulates N-cadherin and matrix metalloproteinase 2 and 9 (*MMP2*, *MMP9*) genes expression and reduces migration and invasion capacity of NSCLC cells [214]. Overexpression of NRF2 in BC cells promotes cell proliferation and metastasis via activating NRF2 target gene *NOTCH1* that in turn induces the expression of genes promoting EMT [215]. NRF2 also positively regulates the *RhoA* gene, which is a critical factor for growth and metastasis, while NRF2 down regulation inhibits proliferation of BC cells [216]. Furthermore, recent findings demonstrate that NRF2 expression is upregulated in human hepatocellular carcinoma (HCC) and that NRF2 promotes proliferation and tumor metastasis by regulating *Bcl-xL* and *MMP-9* genes expression [164]. In contrast, other studies have shown that low expression levels of NRF2 also play a critical role in cancer progression and metastasis formation. In human prostate cancers (PC), NRF2 and its target genes were shown to be significantly decreased during the metastatic process [217]. Additionally, it has also been reported that repression of NRF2 in HCC cell lines increased cell metastasis and invasiveness via TGF-β/Smad-dependent signaling [218]. Moreover, NRF2 deregulation was also shown in other cancers like OC, lung adenocarcinoma (LUAD), human head and neck squamous cell carcinoma (HNSCC) [217,219]. Taken together, all these studies demonstrate that NRF2 has both metastatic and anti-metastatic activity in different types of tumor and stages of cancer progression. It seems like cancer cells utilize both upregulation and downregulation of NRF2 signaling for their advantages. To interfere with cancer metastasis, it will be necessary to fully elucidate the role of NRF2 expression in the metastatic microenvironment.

#### 3.1.5. Metabolic Reprogramming by NRF2: NADPH Links Tumor Growth and Redox Balance

It is well known that cancer cells reprogram their central metabolism to meet the energetic needs imposed by their uncontrolled growth. Initial studies mainly focused on the Warburg effect, while subsequent work also investigated changes in the one-carbon and fatty acids metabolism, pentose phosphate pathway (PPP), tricarboxylic acid cycle (TCA) and glutamine catabolism [173,220,221]. In general, NRF2 can control multiple metabolic routes by two different mechanisms, similarly to other oncogenes (e.g., *MYC* or *KRAS*): the first involves direct transactivation of key metabolic enzymes [222], while the second relies on the modulation of proteins controlling other signaling pathways such as PPARγ [223], Notch [215,224], AHR [225,226] and PI3K/AKT [53]. It is known that alterations of tumor metabolism are often paralleled by an increased antioxidant capacity, which is also part of the adaptive response mounted by cancer cells to face adverse conditions, including OS [173,227,228,229,230]. With this respect, certain metabolic reactions can play a dual role, providing intermediates for biosynthetic processes or essential cofactors used to modulate the intracellular redox balance [231,232,233,234,235]. Among them, NADPH represents a key player in supporting anabolic reactions and ROS-scavenging antioxidant systems. As first, NADPH is essential for the regulation of the glutathione/glutaredoxins (GRXs) system, that regenerates the reduced form of glutathione (GSH) once it is oxidized (GSSG) [236]. Secondly, NADPH is a key cofactor for the glutathione peroxidases (GPXs) that scavenge hydrogen peroxide or potentially harmful alkyl hydroperoxides. Lastly, NADPH is indispensable for the thioredoxin reductases (TRXRs), a class of enzymes that restore the reduced form of thioredoxins (TRXs) and indirectly contribute to the regulation of thiol groups in redox-sensitive proteins [237,238]. Importantly, NRF2 can enhance the expression of genes codying for NADPH-producing enzymes such as *G6PD* (Glucose-6-Phosphate Dehydrogenase) and *PGD* (Phospho Gluconate Dehydrogenase), or enzymes that regenerate glycolytic intermediates that can be diverted into the oxidative PPP branch, such as *TKT* (Transketolase) and *TALDO1* (Transaldolase-1) [53,182,215,239]. Mechanistically, either direct or indirect genes transactivation can occur, depending on the genetic and biological context of NRF2 activation. Indeed, by using H1437, A549 NSCLC and DU145 PC cells, Singh et al., reported that genetic induction of PPP genes was mediated by NRF2-dependent repression of miR-1 and miR-206, two negative regulators of *G6PD*, *PGD* and *TKT* expression, through yet unknown epigenetic mechanisms involving the histone deacetylase HDAC4 [240]. Also miR-1 inhibition was found to underlie NRF2-mediated upregulation of *G6PD* in HCC cells, an event that positively correlated with grading, metastasis number and poor prognosis in HCC patients [241]. Instead, a direct induction of *G6PD*, *PGD*, *TKT*, *TALDO1* and other NADPH-producing enzymes was reported in A549 NSCLC cells with sustained PI3K/AKT pathway activation [53], in agreement with previous ChIP-seq studies [48,182,242]. Also, Xu et al., demonstrated that NRF2 can bind to the AREs within intron 1 and 4 of the *TKT* gene, inducing its expression in HCC cells (SMMC and MHCC97/L) [239]. Consistently, the presence of functional NRF2 binding sites within the AREs of the *TKT* promoter was also reported in MEFs and A549 NSCLC cells, suggesting a direct transactivation mechanism [240]. Also, other data indicate that *NRF2* overexpression or *KEAP1* KD can upregulate the mRNA and protein levels of G6PD and TKT, enhancing tumor proliferation, migration and invasion of MCF7 and MDA-MB-231 BC cells by promoting EMT through G6PD/HIF-1α activation of Notch1 signaling, while *NRF2* silencing or *KEAP1* overexpression reverted these changes [215]. Interestingly, some studies suggest that NRF2 can control the expression of NADPH-producing enzymes involved in one-carbon metabolism such as *MTHFD2* (Methylenetetrahydrofolate Dehydrogenase 2) or in the TCA cycle, such as *IDH1* (Isocitrate Dehydrogenase 1) and *ME1* (Malic enzyme 1) [53,239,243], while others indicate that NRF2-dependent induction of the folate-cycle enzyme *MTHFDL1* (Methylenetetrahydrofolate Dehydrogenase 1-like) can increase the NADPH levels in HCC cells, supporting proliferation and redox homeostasis [58]. 

#### 3.1.6. NRF2 Regulates Metabolic Processes Leading to GSH Synthesis and TCA Cycle Anaplerosis

In line of principle NRF2 can also regulate the redox balance of cancer cells via transcriptional induction of metabolic enzymes or membrane channels that control the availability of cysteine, glutamate and glycine, essential precursors in the GSH synthesis. In this regard, NRF2 was shown to enhance the expression of key enzymes for serine/glycine biosynthesis such as *PHGDH* (Phosphoglycerate Dehydrogenase), *PSAT1* (Phosphoserine Aminotransferase 1), *PSPH* (Phosphoserine Phosphatase) and *SHMT1-2* (Serine Hydroxymethyltransferase 1-2) in a panel of NSCLC cells [167]. Also, a recent work underlined the importance of the NRF2-ATF4 pathway in the regulation of aminoacid metabolism in cancer. Indeed, NRF2 was seen to enhance the ATF4 transcriptional activity in autophagy-deficient HCT116 CRC cells, promoting the expression of genes (*SLC6A9*, *SLC36A4*, *SLC38A1*, and *SLC38A3*) codying for AATs (aminoacid transporters) involved in the uptake of glycine and glutamine. Notably, AATs inhibition sensitized autophagy-deficient CRC cells but not wild-type cells to apoptosis induced by glutamine withdrawal [244]. Other work has focused on xCT, an antiporter that couples the efflux of glutamate with the uptake of cystine (CySS), that is intracellularly reduced to cysteine (2x Cys) by GSH, TRX1 or TRP14 [245,246,247,248,249]. For example, Habib et al., showed that in MCF7 BC cells exposed to OS, an enhanced NRF2 nuclear translocation was responsible for the *SLC7A11* (solute carrier family 7 member 11) gene upregulation, leading to an increase in the xCT mRNA and protein levels and marked glutamate release. Of note, these changes were phenocopied by *NRF2* overexpression or *KEAP1* KD and reverted by its overexpression [250]. In a later study, *NRF2* and *SLC7A11* expression was found to be positively correlated across 947 cancer cell lines from the CCLE dataset [251], especially within 59 different BC cell lines. Here, *NRF2* KD markedly decreased *SLC7A11* expression and glutamate extrusion in Hs578T and MDA-MB-231 BC cells, improving cell viability upon glucose depletion while in the same conditions *NRF2* activation by DMF impaired cell viability. Thus, despite the enhanced activation of the NRF2-xCT axis might efficiently preserve the redox homeostasis of BC cells under OS, it might also decrease their metabolic flexibility, unveiling a specific vulnerability that can be therapeutically exploited [252]. In a study on human melanoma cells with constitutive *BRAF* activation (*BRAFV600E*), Khamari et al., explored potential metabolic changes promoting resistance (A375RIV1 cells) or sensitivity (A375-v cells) to the BRAF inhibitor Vemurafenib, by using an in vivo long-term treated xenograft mouse model. Here, A375RIV1 cells exhibited a strong activation of the NRF2 signaling, followed by an increased expression of genes involved in ROS scavenging (i.e., *GPX1*, *GPX2*), GSH synthesis (i.e., *GCLM,* and *xCT*) and NADPH-generation (i.e., *TKT*, *TALDO1*), compared to the A375-v counterpart. Of note, *NRF2* KD by siRNA decreased the protein content of its target genes in A375RIV1 cells, partly reversing their resistance to Vemurafenib [253]. Thus, the xCT system is key regulator of cancer cells redox balance, while its inactivation might sensitize malignant cells to OS inducers. Of note, xCT overexpression is expected to promote glutamine catabolism to support TCA cycle anaplerosis or GSH synthesis [254]. In this respect, a recent study from Sayin et al., reported that LOF mutations of the *KEAP1* gene can mediate glutamine addiction in both mouse (KPK) and human *KRAS*-driven LUAD cell lines. Here, NRF2 increased activation led to enhanced xCT/*SLC7A11* expression*,* causing an imbalance in the TCA cycle and sensitization of KPK cells to pharmacologic or metabolic glutamine depletion. Importantly, the glutaminase inhibitor CB-839, impaired cell growth in a panel of tumor cells including melanoma, colon, renal, bone, squamous and urinary-tract cancers with *KEAP1* LOF mutations while the use of KI696, a small-molecule activator of NRF2, conversely sensitized *KEAP1 WT* cancer cells previously refractory to CB-839 [255]. Thus, oncogenic alterations in the NRF2-KEAP1 axis can induce defects in central carbon metabolism of cancer cells and reveal metabolic vulnerabilities that can be targeted. Notably, glutamine is the most abundant aminoacid in human serum and is essential for many cancer cells to generate TCA cycle intermediates and support the biosynthesis of nucelotides, N-acetyl glucosamines, fatty acids, GSH and other aminoacids [254]. Intriguingly, NRF2 can control different steps of glutamine fate, from its uptake to its metabolism. For example, early studies reported that NRF2 can induce the expression of the glutamine importer *SLC1A5* in HeLa cells through the ATF4 transcription factor [256] while a recent ChIP-Seq analysis on *KEAP1-/-* mice and human ESCC cells, revealed that NRF2 causes metabolic reprogramming by enhancing the expression of the *SLC1A4* glutamine transporter and other metabolic enzymes [257]. Lastly, the enzyme glutaminase, catalyzing the conversion of glutamine into glutamate, was found to be a direct NRF2 target gene in MCF7 and MCF10 BC cells treated with Sulforaphane or subdued to *KEAP1* KD by siRNA [258]. Therefore, NRF2 is profoundly implicated in the control of glutamine metabolism of cancer cells and most likely this regulatory node will be the focus of extensive research in the next future. 

#### 3.1.7. NRF2 in the Regulation of Fatty Acids Metabolism

Interestingly, NRF2 has been found to exert opposite changes in the metabolism of fatty acids, since a repression of FAS (fatty acid synthesis), but a stimulation of FAO (fatty acid oxidation) has been reported in isolated mitochondria, MEFs (mouse embryonic fibroblasts) and tissues of transgenic mice [259]. Despite the lack of data on malignant tumors a study on HEK-293T cells suggests that NRF2 might control the expression of *CPT1* and *CPT2*, two isoforms of the enzyme carnitine palmitoyltransferase (CPT) that catalyzes the rate-limiting step of FAO [260]. Another mechanism by which NRF2 can potentially support the redox balance of cancer cells is through the suppression of NADPH-consuming processes, including lipid biosynthesis. Indeed, by using murine models with variable *NRF2* expression, Wu et al., showed that *NRF2*-null mice exhibited increased hepatic mRNA levels of the enzymes fatty acid synthase (*FASN*), fatty acid desaturase (*FADS1*, *FADS2*), stearoyl-CoA desaturase (*SCD1*), fatty acid elongases (*ELOVL2*,3,5,*6* and *CYB5R3*), acetyl-CoA carboxylase-1 (*ACC1*) and ATP-citrate lyase (*ACLY*), while the opposite was seen in *KEAP1*-KO mice suggesting that NRF2 restrains lipogenesis and desaturation to prevent NADPH depletion [261]. On the other hand, NRF2 activation in mouse lung was conversely seen to induce the transcription of FAO genes and lipases, promoting degradation of damaged lipids and providing reducing equivalents in the form of NADPH [262]. In conclusion, the activity of NRF2 can significantly affect the efficiency of FAO and lipid biosynthesis, ultimately affecting bioenergetics as well as NADPH-linked antioxidant systems, underscoring the intimate connection between metabolic processes and redox homeostasis.

### 3.2. Strategies to Negatively Modulate NRF2 Signaling/Pathway

Tumors are complex entities composed by heterogeneous cell populations that dynamically adapt to their microenvironment driven by genetic/epigenetic alterations and metabolic rewiring [263,264,265,266,267]. Thus, cancer cell populations that become prevalent during certain phases of the malignant progression can frequently develop resistance to treatment [268,269,270,271]. Moreover, due to patients’ individualities and tumors heterogeneity, variable response rates are often observed making the identification of more effective drugs very urgent [272,273]. In this regard, the concomitant inhibition of antioxidant circuits and metabolic pathways that support the redox balance of malignant cells, delineates a promising anti-cancer strategy [274,275,276,277,278,279,280,281,282,283]. It is known that most of the metabolic changes promoting cancer cells proliferation and tumor growth induce also an increased ROS generation, counterbalanced by an antioxidant response that prevents cell death [234,284,285]. Of note, since the cytotoxicity of conventional anti-cancer therapies largely relies on efficient ROS accumulation, the augmented antioxidant capacity of cancer cells constitutes a key determinant of therapy-resistance [286,287,288]. At the same time, given the strong dependency of malignant cells to their antioxidant systems, interfering with their redox control can induce OS-dependent cell death [289]. Since the NRF2 signaling plays a key role in tumorigenesis, malignant progression [290,291,292] and drug sensitivity [293,294,295], this transcription factor has emerged as a promising therapeutic target [16,282,296,297,298]. In this regard, several studies have tried to identify NRF2 activators to prevent ROS-dependent carcinogenesis while others have focused on the development of NRF2 inhibitors to overcome therapy resistance [299,300,301]. In the following sections, the most promising NRF2 inhibitors, described so far in the literature, will be presented.

#### 3.2.1. Natural Compounds That Impair NRF2 Signaling by Interfering with Protein Synthesis

Despite the increasing demand for negative modulators of NRF2, selective inhibitors are neither currently available nor under clinical trial evaluation. Yet, alternative strategies have been explored based on the pharmacologic manipulation of various NRF2 modulators through natural and synthetic compounds [302,303]. Historically, many natural compounds have been used to treat human pathologies, including cancer. Importantly, these compounds have been often used to potentiate the efficacy of ROS-inducing agents against therapy-resistant cancers, by altering their redox homeostasis. Among the others, many studies focused on Brusatol, a quassinoid from the plant *Brucea javanica* [304,305,306], and its derivatives [307,308]. The antitumor effects of Brusatol have been ascribed both in solid and in hematologic tumors to the inhibition of proliferation and the disruption of antioxidant defenses caused by NRF2 depletion, due to global suppression of protein synthesis [309,310,311]. For instance, in A549 NSCLC cells resistant to radiation, Brusatol was found to dose-dependently decrease the NRF2 protein levels and to enhance the efficacy of ionizing radiations, by inducing ROS-dependent DNA damage and cell death [312]. Also, Brusatol was recently found to impair cell growth and proliferaton of human A375 melanoma cells both in vitro and in vivo (NOD/SCID xenografted mice) when used in combination with UVA treatment, leading to increased ROS generation and apoptosis due to AKT impairment [313]. Also, Karathedath et al., reported that NRF2 overexpression in AML cell lines and primary AML samples caused resistance to Cytarabine, Daunorubicin and Arsenic trioxide (ATO). Importantly, Brusatol was found to markedly decrease the NRF2 content and consequently the expression of its target genes *GCLC*, *GCLM*, *HMOX-1*, and *NQO1*, promoting ROS accumulation and apoptosis [314]. Finally, in a recent study from Xiang et al., the overactivation of NRF2 signaling was found to mediate Gemcitabine resistance in human PANC-1, PATU-8988 and BXPC3 pancreatic ductal adenocarcinoma cancer (PDAC) cell lines. Here, Brusatol induced ROS accumulation, growth inhibition and apoptosis by decreasing the protein levels of NRF2 and its target genes *HO-1* and *NQO-1*. Notably*,* Brusatol enhanced the antitumor activity of Gemcitabine in vivo by reducing the growth rate of PANC-1 xenografts implanted in BALB/c nude mice [315]. Similarly, other work has been directed to Halofuginone from the plant *Dichroa febrifuga*, which is rapidly emerging as one of the most promising NRF2 inhibitors. Indeed Halofuginone has been shown to impair proliferation, migration and invasion of HepG2 HCC cells [316] and MCF7 BC cells [317,318], to reverse the radioresistance of Lewis lung cancer cells (LLCC) [319], or to improve the drug delivery in PDAC cells [320]. In a recent study from Tsuchida K. et al., Halofuginone was seen to repress protein synthesis through prolyl-tRNA synthetase inhibition and to prevent NRF2 nuclear accumulation causing the impaired expression of enzymes involved in drug metabolism and transport, iron metabolism, GSH metabolism and ROS scavenging. As a result, KYSE70 human esophageal cancer (HEC) or A549 NSCLC cells, became more susceptible to Cisplatin based on in vitro and in vivo experiments, with limited side-effects [321]. It should be emphasized that Halofuginone has a very similar mechanism of action to that of Brusatol, confirming that the inhibition of protein synthesis remains a valid strategy to block NRF2 signaling. 

#### 3.2.2. Natural Compounds That Impair NRF2 Signaling by Acting on Functional Regulators

Other works have explored the potential use of Chrysin, a natural flavone found in many plant extracts such as honey, propolis, and blue passion flowers, endowed with anticancer effects against different tumors [322,323]. In an earlier study on BEL-7402 human HCC cells resistant to Doxorubicin, Chrysin significantly reduced NRF2 expression both at the mRNA and protein levels, by interfering with PI3K/AKT and ERK pathways. As a result, the expression of NRF2-related target genes *HMOX-1*, *AKR1B10* and *MRP5*, was significantly reduced and the chemoresistance attenuated [324]. In another work form Wang J. et al., Chrysin was seen to suppress the proliferation, migration and invasion of human T98, U251 and U87 glioblastoma (GBM) cells and to abrogate the in vivo tumorigenicity of U87 xenografts in BALB/c mice. Mechanistically, Chrysin impaired NRF2 nuclear translocation, by decreasing the protein levels of phospho-extracellular signal-regulated kinase-1/2 (ERK1/2) and two antioxidant enzymes HO-1 and NQO-1 [325]. 

#### 3.2.3. Natural and Synthetic Compounds Blocking NRF2 Pathway by Still Unknown Mechanisms

Oridonin, a diterpenoid isolated from the herb *Rabdosia rubescens*, represents another promising compound whose potent antitumor effects have been described in leukemia [326], BC [327] and CRC cells [328]. Very recently, Oridonin was found to promote mitochondrial-dependent apoptosis by activating PPARγ and suppressing both NF-κB and NRF2 pathways in MG-63 and HOS osteosarcoma cells. Here, Oridonin prevented NRF2 nuclear translocation and decreased the expression of the *HMOX1*, *NQO1* genes and their encoded antioxidant proteins, leading to ROS-dependent apoptosis, in vitro and in vivo [329]. Also Plumbagin, a natural naphthoquinone from *Plumbago zeylanica L.*, has received substantial attention as a potent inducer of apoptosis in pancreatic, lung, breast and prostate cancer cells caused by ROS overproduction [114,330]. With this respect, by using human ovarian (OVCAR3), breast (SKOV3, MCF7), and endometrial (ECC1) cancer cells, a recent study has shown that Plumbagin promotes ROS generation via the mETC complexes I-III and synergizes with Brusatol to block NRF2 pathway, triggering cell death [331]. In another work, a SILAC-based quantitative proteomic approach was used to characterize the biological changes induced by Plumbagin in SCC25 tongue squamous cell carcinoma (TSCC) cells. Here, Plumbagin was seen to decrease the NRF2 nuclear translocation and to suppress the induction of its target genes. As a result, unbalanced ROS overproduction led to cell cycle arrest and stemness attenuation, triggering apoptosis [332]. Extensive research has also been pursued on the alkaloid Trigonelline, a coffee extract initially identified as a negative modulator of NRF2 signaling in HT29 human colorectal cancer cells (CRC), with the ability to decrease the nuclear and total content of NRF2 and to inhibit its target genes transactivation [333]. In a later study, the effects of Trigonelline on NRF2-mediated apoptosis evasion were studied in MiaPaca2, PANC-1, and Colo357 PDAC cell lines and in the human pancreatic duct cell line H6c7. Here, Trigonelline was shown to strongly suppress NRF2 activity by preventing its nuclear accumulation and to increase the efficacy of TRAIL and Etoposide both in vitro and in vivo, with few side-toxicity [334]. Importantly, three different studies from the group of Shin D. recently focused on the impact of Trigonelline on chemoresistance by using experimental models of HNC. First, using HNC cells resistant to Cisplatin, Trigonelline was found to restore the chemosensitivity of HN 2-10 and SNU cells both in vitro and in vivo when combined with inhibitors of GSH synthesis and TRX system [287]. Interestingly, in the same experimental model, NRF2 was seen to promote resistance to the ferroptosis inducer Artesunate, while conversely the combination with Trigonelline resulted in strong cytotoxicity due to ROS accumulation. Remarkably, the effect was restricted to Artesunate- or Cisplatin-resistant HNC cells, sparing normal oral keratinocytes and oral fibroblasts [301]. Lately, the same group reported that in HNC cells, suppression of NRF2 signaling by Trigonelline could reverse the resistance to ferroptosis both in vitro and in vivo [335]. Thus, Trigonelline is emerging as a promising molecule for combination regimens against tumors with widespread chemoresistance. Importantly, the use of high throughput screening (HTS) has played a major role in the discovery of NRF2 inhibitors. For example, AEM1, a benzodioxole substituted analog recently identified in A549 NSCLC cells, was seen to strongly suppress NRF2-mediated genes expression, without altering NRF2 protein stability, its phosphorylation or the KEAP1 content. Importantly, AEM1 reduced the growth rate of A549 NSCLC cells both in vitro and in vivo, enhancing also their sensitivity to Etoposide and 5-Fluorouracil. Of note, AEM1 was also able to decrease the NRF2-dependent induction of *HMOX1* in H838 and H460 NSCLC cells with LOF mutations of the *KEAP1* gene, suggesting that AEM1 might preferentially target tumor cells with constitutive NRF2 activation [336]. Similarly, Singh et al., conducted a quantitative high-throughput screening (qHTS) of the Molecular Libraries Small Molecule Repository (MLSMR) and identified a compound named ML385 that was able to reduce the transcriptional activity of NRF2 by preventing the binding of the NRF2-MAFG complex to the ARE sequence in the promoter of NSCLC cells. More in detail, ML385 attenuated NRF2 pathway by affecting the DNA binding activity of the NRF2-MAFG protein complex through direct interaction with NRF2. As a result, ML385 induced selective toxicity in both A549 and H460 NSCLC cells harboring *KEAP1* mutations, enhancing the cytotoxic effects of Doxorubicin, Carboplatin and Paclitaxel without affecting the non-tumorigenic BEAS2B cells. Importantly, the therapeutic efficacy of ML385 as a single agent and in combination with carboplatin was also confirmed in orthotopic lung cancer xenografts subcutaneously implanted in nude mice [25]. Moreover, in a very recent study from Hori R. et al., the K-563 compound was isolated from *Streptomyces sp*. after HTS screening of almost 10000 culture broth samples. Of note, K-563 abrogated the expression of NRF2 target genes such as *GCLC*, *GCLM*, *AKR1C1* (reductase family 1 member C1), *ME1*, *NQO1* and *TXNRD1* in A549 NSCLC cells and in the human gallbladder cancer (GBC) cell line TGBC24TKB, without altering the NRF2 protein levels, its ARE-binding ability, or its nuclear localization. As a result, K-563 promoted ROS accumulation and synergized with either Cisplatin or Etoposide in A549 NSCLC cells, suppressing also cell proliferation in GBC cells (TGBC24TKB) without affecting normal human lung cells (BEAS-2B) [337]. In the context of hematologic tumors, Zhang et al., identified 4f, a pyrazolyl hydroxamic acid derivative with potent antineoplastic effects in human THP-1, HL60 and U937 AML (acute myeloid leukemia) cells. Here, 4f was found to decrease the NRF2 protein content and the mRNA levels of *HMOX1* and *GCLC* genes, promoting increased caspase-3 and PARP cleavage. Besides, 4f suppressed the growth of THP-1 xenografts seeded onto the CAMs of chicken eggs and also impaired blood vessels formation in vivo in a gelatin sponge assay, suggesting that 4f might be a promising treatment in advanced AML [338]. In summary, very promising results have been reported in case of repressors of protein synthesis, an evidence that might pave the way to the design of novel strategies to target the NRF2 pathway in cancers. Taken together, these studies provide a strong rationale for the identification and validation of compounds with the ability to disrupt the redox control exerted by NRF2 in solid as well as hematologic tumors. Moreover, it is expected that other experimental work as well as refined clinical trials will better define the molecular mechanisms, the specific context and the types of tumors wherein this approach might ensure the optimal efficacy in patients with advanced cancers.

#### 3.2.4. Natural and Synthetic Compounds Targeting Functional Regulators of the NRF2-KEAP1 Pathway

An alternative strategy to hamper the pro-tumoral effects of NRF2 is through the modulation of functional regulators that ultimately converge on this signaling pathway. This approach offers at least two significant advantages. First, it relies on already established drugs with proven anticancer activity and, more importantly, it does not require selective NRF2 inhibitors. Of note, recent work supports the applicability of this strategy. For example, LGK-974, a specific inhibitor of the O-acyltransferase porcupine (PORCN) used to disrupt the WNT signaling, was recently found to prevent NRF2 nuclear translocation and its expression in HepG2 cells, presumably by impairing the WNT3A-dependent formation of the GSK-3/β-TrCP protein complex. As a result, the HepG2 cells were sensitized to otherwise non-toxic radiation doses, due to downregulation of NRF2 target genes such as *HMOX1* and *NQO1* and increased ROS production [339]. Also Wogonin, a flavonoid isolated from *Scutellaria baicalensis Georgi*, has emerged as a promising anticancer agent, due to its chemosensitizing ability in Doxocycline-resistant MCF7 BC cells [340]. Further investigations revealed that Wogonin reduced the NRF2 nuclear translocation and promoted increased ROS production, potentiating the cytotoxicity of Hydroxy-Camptothecin, Cisplatin and Etoposide also in HepG2 cells [341]. Other work was conducted on chronic myelogenous leukemia (CML) cells sensitive (K562) or resistant (K562/AO2) to Adriamycin (ADR). As first, Wogonin was found to decrease the NRF2 mRNA and protein levels in CML cells, causing a marked reduction in the HO-1, NQO1 and MRP1 proteins [342]. In a later work the same authors showed that Wogonin could decrease the binding of both p65 and p50 NF-kB subunits to the NRF2 promoter in K562/A02 cells, enhancing their sensitivity to ADR. In vivo, transplantation experiments of K562/A02 cells into NOD/SCID mice proved that the combination of ADR and Wogonin could reduce the nuclear content of NF-κB p65 and NRF2 [343]. Hence, Wogonin represents a promising inhibitor of NRF2 and a potent chemosensitizer in solid and hemathologic tumors. Also, Gao et al., explored the anti-cancer potential of Apigenin, a natural bioflavonoid found in many fruits and vegetables. As first, Apigenin was seen to reduce the expression of NRF2 and its targets HO-1, AKR1B10 and MRP5 both at the mRNA and protein levels in a KEAP1-independent way, by downregulating the PI3K/AKT pathway and to strongly sensitize BEL-7402/ADM cells to Doxorubicin both in vitro and in vivo, blocking tumor growth [344]. Additional work, led to the identification of the microRNA mir-101, as a modulator of NRF2 functions in HCC cells. Here, mir-101 was found to be downregulated in BEL-7402/ADM (resistant) and conversely highly expressed in BEL-7402 (sensitive) cells, while the NRF2 mRNA sequence analysis revealed the presence of a complementary site for mir-101 in the 3’-UTR. Importantly, mir-101 mimics markedly decreased the NRF2 protein levels in BEL-7402/ADM cells exerting chemo-sensitizing effects, while opposite changes were induced by antimir-101. Collectively, these data indicate that mir-101 re-expression might be a novel mechanism to blunt NRF2 signaling in HCC resistant cells [345]. Also, Retinoic acid, a Vitamin A metabolite, has emerged as a potential anticancer agent inducing differentiation, growth arrest, and apoptosis of cancer cells [346]. Initial studies on solid tumors have shown that All-*Trans* Retinoic Acid (ATRA) can induce RAR*α* expression, which in turn forms a protein complex with NRF2 and antagonizes its transactivation [347,348], sensitizing chemoresistant neuroblastoma (NB) HTLA-230 cells to the proteasome inhibitor Bortezomib [349]. Other experimental work from Valenzuela M. et al., extended these observations to AML and APL cells treated with the ROS-inducer Arsenic Trioxide (ATO). Here, the combined use of ATRA and ATO was seen to prevent NRF2 nuclear translocation and to suppress the antioxidant response in HL-60 and THP-1 AML and in NB4 APL cells, inducing cytotoxicity. Additionally, the authors also showed that the ATRA-mediated inhibition of NRF2 depended on RAR*α*, since ATRA was ineffective when parental NB4 cells were pre-treated with the RAR*α* antagonist Ro-41-5253 or when applied to mutant NB4-R2 cells lacking RAR*α* expression [350].

## 4. Conclusions and Future Perspectives

Since its discovery in 1994 [27] NRF2 has been connected to various cellular mechanisms, such as response to oxidative stress, mitochondrial respiration, stem cell quiescence, mRNA translation and autophagy [51]. The NRF2-KEAP1 pathway is a master regulator of cell protection mechanisms against exogenous or endogenous stress sources. Thus, in the past decade, NRF2 has emerged as an important target in cancer therapy. Many questions have arisen in relation to risks and benefits of negative modulators of NRF2 signaling. Recent research suggest that suppression of antioxidant mechanisms involving NRF2 might potentially induce a pro-oxidizing shift in tumor microenvironment and promote ROS-dependent cell death in many cancer types. Despite the absence of specific and selective NRF2 inhibitors, convincing indications show that the use of natural compounds with known therapeutic action might be effectively used in different types of tumors [351,352]. NRF2 has been recognized as one of the critical factors regulating an array of genes that protect cells against xenobiotics. NRF2-mediated transcriptional regulation is coordinated by a number of specific events within the cellular environment. Among them, the triggering stimulus, the cooperation with other activators and repressors, the interplay with different signaling pathways and the epigenetic landscape of the target gene promoters, can be regarded of utmost importance. Many approaches have been devised to target the NRF2 signaling pathway in cancer such as regulating NRF2 expression at the transcriptional level, controlling the NRF2 nuclear translocation, targeting the KEAP1-NRF2 binding for modulating NRF2 protein stability and regulating NRF2 binding to its target genes promoters. Several small-molecule NRF2 activators and inhibitors have been developed and successfully employed in cancer treatment. The important drawbacks of targeted therapy include resistance of cancer cells and difficulties of developing drugs to some tumor-specific targets [191,353]. Studies suggest that in tumors, high levels of NRF2 can occur in absence of genomic alterations in the NRF2 and KEAP1 genes. Additional research will help to increase the specificity of NRF2-based therapies. For example, crystallographic investigations of KEAP1 might provide chances to design and synthesize molecules that selectively interfere with the KEAP1-NRF2 binding [354]. Moreover, accumulating evidence suggests that NRF2 can interact with other pathways implicated in cell survival [355]. These important points offer great opportunities for pharmacological intervention to control the level of NRF2 and its therapeutic effects. NRF2 was firstly described as a tumor suppressor involved in the inhibition of tumor initiation and cancer metastasis. Recent evidence showed that it can also act as a pro-oncogenic factor. It is now well accepted that NRF2 has a dual role in carcinogenesis and that pharmacological induction of the NRF2 pathway might be chemopreventive in the early stages of tumorigenesis. However, adverse effects might occur in advanced stages of cancer by inducing therapy resistance [356,357]. Chemotherapy resistance is one of the crucial problems for the effective treatment of many cancers, and NRF2 inhibition might be a promising approach to overcome this phenomenon. Being located at the crossroad of many defensive pathways influencing cell life during chemical, oxidative, and metabolic stress, the NRF2-KEAP1 pathway has been the focus of broad research aimed at revealing its role in cancer [105]. Aberrant activation of the NRF2-KEAP1 pathway is often recognized in many tumors, promoting cancer growth, survival, metastasis formation and therapy resistance [358,359]. It is known that NRF2 activation in response to oxidative stress promotes cells survival. NRF2 also induces metabolic changes that sustain cell proliferation. Because of these benefits, cancer cells with sustained NRF2 activation frequently develop “NRF2 addiction” [3]. Disruption of NRF2 signaling is a promising therapeutic approach against NRF2-addicted cancers and some effective NRF2 inhibitors such as brusatol and halofuginone have been employed, while many others are under investigation. Nevertheless, administration of systemic NRF2 inhibitors may cause off-target effects on patients and this represents a key aspect for the identification of novel compounds that should ideally possess high specificity, bioactivity and limited side-toxicity [26]. To prevent these side effects, mechanistic insights should be revealed and new pharmacological targets besides NRF2 should be studied. Depending on metabolic fluxes of NRF2-addicted cancer cells, thorough metabolite examination might help to identify specific diagnostic markers. Research have suggested that the over-activation of NRF2 promotes tumor development, prevents cell apoptosis and enhances the therapy-resistance of cancer cells. We know that both NRF2 inducers and NRF2 inhibitors possess anticancer activity against different targets. NRF2 inducers protect normal cells from carcinogen effects, while NRF2 inhibitors halt cancer cells proliferation. The question is what we need exactly, NRF2 activators or inhibitors? According to the numerous recent studies related to the oncogenic activity of NRF2, the synthesis of novel selective inhibitors of NRF2-KEAP1 pathway could be the better strategy for cancer prevention and therapy [64]. The dilemma of whether NRF2 can be used as a pharmacological target needs to be harmonized with the research of clinical application and participation of mediators of oxidative stress in the anticancer therapy. In conclusion, the discovery, design, and synthesis of NRF2-centered methodologies are important and challenging tasks that might pave the way to novel therapeutic approaches in cancer treatment.

**Table 2 antioxidants-09-00193-t002:** Systematic presentation of studies on possible use of natural Nrf2 activators in therapy with possible mechanisms of action connected to Nrf2 activation pathways.

Natural Products					
Type of Study	Experimental Model	Treatment Doses and Duration	Observed Mechanism of Action/Effects	Proposed Application in Therapy	Ref.
In vivo	male Albino rats Wistar strain	200 mg/kg dose of curcumin for four consecutive days oral administration	enhanced nuclear translocation and ARE-binding of Nrf2 curcumin protects against dimethylnitrosamine (DMN)-induced hepatic injury	chemopreventive agent	[360]
In vivo	male C57BL/6J mice	daily treated with curcumin at the dose of 50 mg/kg body weight by oral gavage	mediated nuclear translocation of Nrf2, followed by induction of HO-1 curcumin intervention dramatically reversed the defects in Nrf2 signaling induced by high fat diet	improving glucose intolerance	[361]
In vivo	male Sprague–Dawley rats	supplemented with curcumin (1 g/kg diet) for 16 weeks	increased HO-1 expression was increased along with suppressed oxidative stress as well as reduced hepatic fat accumulation and fibrotic changes	attenuating the pathogenesis of fatty liver induced metabolic diseases	[362]
In vivo/In vitro	female specific pathogen-free BALB/c mice mouse macrophage RAW264.7 cells	on day 22, the mice were treated with curcumin (200 mg/kg) 1 h before ovalbumin challenge. cells treated with different concentrations (0, 5, 10, 25, and 50 μmol/L) of curcumin for 24 h or with 50-μmol/L curcumin for different lengths of time (0, 4, 8, 12 and 24 h)	Nrf2 and HO-1 levels in lung tissues were significantly increased heme oxygenase-1 and nuclear Nrf2 levels were enhanced in dose- and time-dependent manners in curcumin-treated RAW264.7 cells.	potentially effective drug in asthma treatment (alleviate airway inflammation in asthma through the Nrf2/HO-1 pathway)	[363]
In vitro	primary cultures of cerebellar granule neurons(CGNs) of rats	pretreated with 5–30 μM curcumin	mediated neuro-protection against hemin induced damage via Nrf2 dependent HO-1 expression	neuroprotective agent	[364]
In vitro	human breast cancer cell line MCF-7	treatment with DMSO (vehicle) or various concentrations of curcumin for 12, 24 or 48 h	inhibition of the proliferation of breast cancer cells through Nrf2-mediated down-regulation of Fen1 expression	chemotherapeutic agent	[365]
In vivo	female Sprague-Dawley rats	food supplemented with resveratrol equivalent to 50, 100, or 300 mg/kg body weight/d	elevated protein and mRNA expression of hepatic Nrf2, attenuation of oxidative stress and suppression of inflammatory response mediated by Nrf2	prevention and intervention of human hepatocellular carcinoma	[366]
In vivo/In vitro	female August Copenhagen Irish rats non-tumorigenic human breast epithelial cell line MCF-10A	resveratrol given as a 50 mg subcutaneous pellet every other month during 8 months to animals subcutaneously treated with 3 mg E2 pellets prepared in cholesterol cells were treated with E2 (10 and 50 nM) and Res (50 µM) for up to 48 h	resveratrol treatment alone or in combination with E2 significantly upregulated expression of nuclear factor erythroid 2-related factor 2 (NRF2) in mammary tissues; increased expression of NRF2-regulated genes codying for antioxidant enzymes (NQO1, SOD3 and OGG1) involved in protection against oxidative DNA damage inhibition of suppression of FMO1 and AOX1 genes expression via regulation of NRF2	a potential chemopreventive agent in the development of therapeutic strategies for the prevention of estrogen-induced breast neoplasia	[367]
In vitro	MCF-10F and MCF-7 cells	retreated with 0.1 to 30 nmol/L TCDD with or without 25 μmol/L resveratrol for 72 h and then incubated with E2 (0.1–10 μmol/L) for 24 h	induction of NAD(P)H quinone oxidoreductase 1 (NQO1) in MCF-10F cells exposed to resveratrol may involve the Nrf2-Keap1-ARE pathway (dissociation of Nrf2 from Keap 1 and translocation of Nrf2 to the nucleus, where it binds to the ARE to activate the transcription of NQO1 mRNA)	a potential chemopreventive agent against estrogen-initiated breast cancer	[368]
In vitro	primary rat hepatocytes were obtained from Sprague–Dawley male rats	cells were incubated in the presence of resveratrol for 24 and 48 h ( at concentrations of 25, 50 and 75 µM)	increase in the level of Nrf2 and induction of its translocation to the nucleus, and the increase in the concentration of the coding mRNA for Nrf2	protection of liver cells from oxidative stress induced damage (chemopreventive agents)	[115]
In vitro	human type II alveolar epithelial cell line, A549	cells were treated with various concentrations of native EGCG (5, 10, 20, 40, 60, 80 and 100 μM) and nano EGCG (1, 2, 4, 6, 8, 10 and 12 μM) and allowed to grow for 48 h	increased expression of Nrf2 and Keap1 in native and nano EGCG treated A549 cells that could regulate apoptosis	chemotherapeutic in lung cancer	[369]
In vitro	bovine aortic endothelial cells (BAECs)	cells treated with various concentrations of EGCG	upregulates Nrf2 levels in nuclear extracts and increases ARE-luciferase activity	therapeutic targets in a variety of oxidant- and inflammatory-mediated vascular diseases	[370]
In vivo	male albino Wistar rats; animal model of bleomycin-induced pulmonary fibrosis	intraperitoneally treated with EGCG at a dosage of 20 mg/kg body weight, once daily throughout 28 days	EGCG enhances antioxidant activities and Phase II enzymes involving Nrf2–Keap1 signaling, with subsequent restraint inflammation during bleomycin-induced pulmonary fibrosis no significant change in the expression of Keap1	treatment of diseases such as pulmonary fibrosis	[371]
In vitro	human hepatocytes (HHL5) and hepatoma (HepG2) cells	exposed to various concentrations of sulforaphane for different times with DMSO (0.1%) as control	increased nuclear Nrf2 levels and intracellular GSH levels in both cell lines but with slightly different pattern indicating a potential risk of chemo-resistance of using sulforaphane for chemoprevention	possible induction of pro-survival effects in cancer cells	[128]
In vivo	male BALB/c mice (6 weeks)	5 μmol/animal sulforaphane plus different doses of microcystin-LR	Nrf2 translocation to the nucleus in mouse livers induction of Nrf2 downstream target genes, NQO1 and HO-1, two phase II enzymes	effective in cytoprotection against MC-LR-induced hepatotoxicity	[372]
In vitro	adult rat cardiomyocytes	exposed to 5μM sulforaphane with or without H_2_O_2_	upregulation of Nrf2 by sulforaphane occurs at 1 h of incubation increased protein expression of PGC-1α	protective action against oxidative damage, however, timeline of the sulforaphane actions needs to be established	[373]
In vivo	male BALB/c mice (6 weeks)	5 μmol/animal sulforaphane plus different doses of microcystin-LR	Nrf2 translocation to the nucleus in mouse livers induction of Nrf2 downstream target genes, NQO1 and HO-1, two phase II enzymes	effective in cytoprotection against MC-LR-induced hepatotoxicity	[372]
Electrophilic/Covalent					
Triterpenoids					
In vitro	K562 myeloid leukemia cells	Exposed to 0.05–10 μM CDDO-Me for 24–48 h	Down-regulated Na+/K+ ATPase Arrested cells in G2/M and S phases Increased mitochondria and death receptor-dependent and ER-mediated apoptosis Triggered activation of autophagy	Activated and potentiated the effects of the apoptosis and autophagy pathways to kill K562 cancer cells	[374]
In vitro and In vivo	SKOV3, OVCAR3, A2780, A2780/CP70 and Hey2 ovarian cancer cells	0–50 μM CDDO-Me depending on cell assays 20 mg/kg CDDO-Me in xenograft model using nude mice	Binds to USP7 cells, decreasing MDM2, MDMX and UHRF1 Suppresses tumor growth in a xenograft model	Targets apoptosis-related substrates, increasing apoptosis and reducing growth of ovarian cancer cells	[375]
In vitro	MiaPaCa-2 and BxPC-3 cell lines 6 week old Scid/Ner mice	0.625–5μM CDDO-Me in cell culture CDDO-Me 7.5 mg/kg × 5 days/wk by oral gavage until day 40 (to treat primary tumor) or day 100 (to treat residual disease)	Decreased proliferation and increased apoptosis in K-ras normal and mutant cells Inhibits antiapoptotic pathways Inhibited tumor growth in mice Increased survival time of mice	Combination of in vitro and in vivo effects demonstrate that CDDO-Me will increase apoptosis in pancreatic ductal adenoma carcinoma cell lines and improve the survival of animals	[376]
In vitro	MDA-MB 435, MDA-MB 231, MDA-MB 468, BT-549, T47D and MCF-7 breast cancer cells	CDDO-Me 1.5 μM for 4 h	Induces endoplasmic reticulum vacuolation Induces apoptotic pathways Increases intracellular calcium and generation of reactive oxygen species	CDDO-Me-induced c-FLIPL downregulation and relationship between Ca2+ influx and ROS generation are keys in controlling breast cancer growth.	[377]
In vitro	HO8910 and SKOV3 ovarian cancer cells	CDDO-Me concentration range of 0–100 μM 5 μM CDDO-Me in assays requiring a single concentration	Upregulates Hsp70 Degrades Hsp90-associated protein—ErbB2 and Akt CDDO-Me reacts with nucleophiles on Hsp90 to form Michael adducts	May provide added insight to CDDO-Me action, with Hsp90 as a novel target	[378]
In vivo	C57BL/6 WT mice LSL-KrasG12D/+; Pdx-1-Cre (KC) mice Polyoma-middle T (PyMT) mice	CDDO-Im (100 mg/kg diet) fed 2 days prior to LPS injections	Reduced the lethal effects of LPS injection in mutant mice Increased migration of CD45+ cells Increased percentage of Gr1+ myeloid-derived suppressor cells CDDO-Im decreased IL-6, CCL-2, VEGF, and G-CSF in mutant mice	Postulating the use of CDDO-Im as prophylaxes in the development of pancreatic cancer within susceptible populations. This is due to the reduction in proinflammatory mediators.	[379]
In vitro	Human Jurkat E6-1 cells	CDDO-Im 1 nM and 10 nM for 30 min prior to activation with anti-CD3/anti-CD28	Inhibited production of IL-2 Suppression of CD25 in Nrf2-dependent manner with no effect on CD69	Nrf2 activation by CDDO-Im reduces IL-2 secretion and CD25 expression suggesting a role in potential anticancer therapy.	[380]
Dithiolethiones			∙		
In vitro	Mouse carcinoma Hepa-1c1c7 cells	Oltipraz 5–60 μM anetholedithione (ADT) 3–15 μM 1,2-dithiole-3-thione (D3T) 1–5 μM	Dithiolethiones are reduced in Hepa-1c1c7 cells leading to generation of superoxide radicals Superoxide dismutates to form hydrogen peroxide promoting translocation of Nrf2 to nucleus Translocated Nrf2 results in the upregulation of various Phase II enzyme expression.	Use of D3T and members of this family may be able to modify KEAP-1 activity and upregulate the expression of Phase II enzymes	[381]
In vivo	Male Fischer 344 (100 g)	D3T administered by oral gavage 0.5 mmol/kg (in distilled water with 1% Cremaphor, and 25% glycerol	Increased heme oxygenase (HO-1) activity Increased expression of HO-1 Increased levels of ferritin	Very early paper describing the potential utility of Dithiolethiones, like D3T, may offer protection against pro-carcinogenic compounds that increase oxidative stress	[382]
In vitro	HT29 colon adenocarcinoma cells	30μM D3T in DMSO vehicle (less than 0.1% final DMSO concentration)	Increased expression and activity of multiple reductases—Thioredoxin reductase 1, prostaglandin reductase 1, NAD(P)Quinone oxidoreductase 1 Potentiation of hydroxymethylacylfulvene action in alkylating DNA	D3T was a much stronger inducer of reductases compared to selenite and increased potency of the anticancer drug, hydroxymethylacylfulvene	[383]
In vivo	Male Fisher 344 rats (90–100 g)	Oral gavage of DST at 0.03 to 0.3 mmol/kg body wt at 3 days/week for 3 weeks Also 0.1 mmol/kg for measuring hepatic protein expression	Reduced the pre-cancer potency of aflatoxin B by inhibiting the expression of glutathione S-transferase-placental isoform Blocked aflatoxin-mediated increase in Induced multiple hepatic genes associated with detoxifying aflatoxin—including glutathione S-transferase A5 (GSTA5) and AFB1 aldehyde reductase	D3T is more potent than older Dithiolethiones like oltipraz and could be a probe for measuring anticancer potencies of this drug class	[384]
In vitro	HepG2 hepatic and LS180 colon cells	S-diclofenac and S-sulindac in range of 0–100 μM	Inhibited activity and expression of CYP1A1, 1B1 and 1A2 Regulates aryl hydrocarbon receptor pathway Blocked binding of aryl hydrocarbon to the responsive element Increased expression of anticancer enzymes, glutathione S transferase, glutamate cysteine ligase, and glutathione reductase	The NSAIDs with the dithiolethione group, S-diclofenac and S-sulindac, may function as effective anticancer agents	[385]
In vitro	HT29 and HCT116 colon adenocarcinoma	100 μM Oltipraz with 0.2% MeSO as a solvent control	Induction of p65, IκB kinase α (IKKα), IκB kinase β (IKKβ), and NF-κB-inducing kinase (NIK) IκBα phosphorylation was also induce Induction of protein binding to a consensus NF-κB element Transcriptional activation of QR was decreased Both MEKK3 and NIK exert effects on IKKα/β activation via different pathways	This is a novel pathway involved in QR gene regulation and may provide insight to the actions of oltipraz as an anticancer agent	[386]
**Non-Electrophilic/Non-Covalent**					
In vivo and in vitro	Male C57/Bl6 mice (22 g) and bone marrow-derived mouse macrophage cells	RA839 was dissolved in a vehicle of 95% (*v*/*v*) hydroxyethyl cellulose (0.5% (*w*/*v*))/5% (*v*/*v*) solutol—injected IP at 30 mg/kg. General RA839 interactions measured at a concentration of 10 μM	Binds to KELCH domain with affinity of 6 μM Blocks Nrf2-Keap-1 protein-protein interaction Modified activity of multiple genes in mouse macrophage cells Reduced the induction of nitric oxide release and inducible nitric oxide synthase activity Induced Nrf2 gene expression in mouse liver	RA839 may be a useful tool in developing anticancer drugs that target the prevention of Nrf2-Keap1 protein interaction.	[80]
In vitro	THP-1 cells	Cells were exposed to TAT14 using a concentration range of 0–75 μM	Activates heme oxygenase 1 expression and activity Did not alter Nrf2 mRNA expression Reduced LPS-induced TNF expression	The 14-mer TAT fragment interacts with Keap1, preventing Nrf2-Keap1 association, allowing Nrf2 to activate mediators downstream	[387]
In vitro	HepG2 hepatic and U2OS bone cell lines	ML334 and its isomers in a concentration range of 0–100 μM	Binds with high affinity to KEAP1 Strong inducer of ARE activity in both hepatic and bone cell lines	First description of ML334 as a potent inhibitor of Nrf2-Keap1 interaction. Highly potent.	[388]
In vitro	Immortalized baby mouse kidney epithelial cells (iBMK) and MDA-MB-231 breast cancer cells	Geopyxin F and other “geopyxin” isomers were used in a concentration range of 0–70 μM depending on assay	Geopyxin F was the most potent of Geopyxin compounds at inducing Nrf2 activity Geopyxin F displayed almost not toxicity/lethality in MDA-MB-231 cells Geopyxin F increased autophagosome formation in iBMK cells Geopyxin F prevents ubiquitination and stabilizes Nrf2 (KEAP1-dependent)—but is independent of interactions at the Cys151 in KEAP1	Geopyxin F, demonstrated a higher level of protection compared to electrophilic Nrf2 activators. Heightened potency suggests that Geopyxin F may be a useful anticancer compound.	[78]
In vitro	MCF-7 breast cancer cells	Multiple drugs were used as “off-label” activators of Nrf2 Astemizole 8 μM Tamoxifen 1 μM Trifluoperazine 10 μM	Astemizole, Tamoxifen and Trifluoperazine increased expression of the NQO1, HO-1, and GCLM genes After 24 h, the stimulated gene expression to basal was highest in Astemizole compared to sulforaphane The genes for the detoxifying enzymes—NAD(P)H quinone oxidoreductase 1 (NQO1), heme oxygenase 1 (HO1) where most sensitive to upregulation	After large-scale screening, select drugs were chosen based on their ability to activate Nrf2. This shows that ‘off-label’ mechanisms may have benefit as anticancer drugs. Astemizole was the best candidate.	[79]

## Figures and Tables

**Figure 1 antioxidants-09-00193-f001:**
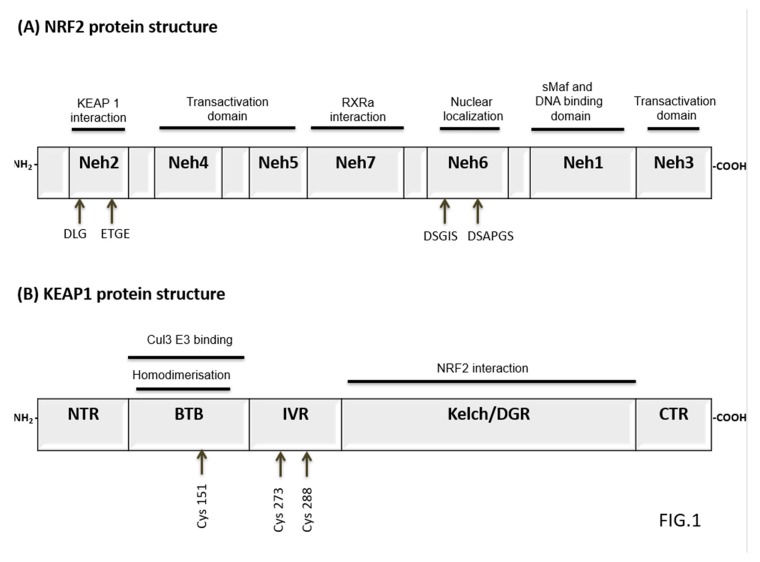
Domain architectures of Kelch-like ECH-associated protein 1 (KEAP1) and nuclear factor erythroid 2-related factor 2 (NRF2). (**A**) Human NRF2 protein is 605 amino acids long and contains seven Neh domains. The Neh1 contains a basic leucine zipper motif that is responsible for dimerization with sMaf protein and ARE sequence binding in DNA. Neh2 has two binding sites known as DLG and ETGE that control KEAP1 interaction. The Neh6 domain is a serine-rich domain containing two motifs (DSGIS and DSAPGS) that negatively regulate NRF2 stability. The Neh7 domain interacts with a nuclear receptor RXRα. The Neh3, Neh4, and Neh5 domains are known as trans-activation domains of NRF2. (**B**) KEAP1 is a 69-kDa protein and contains five domains. The BTB domain is critical for KEAP1 dimerization and recruitment of Cul3-based E3-ligase. The IVR domain has hypercritical cysteine residues, Cys273 and Cys288 that are essential for controlling NRF2 activity. Kelch/DGR domain negatively regulates NRF2 activation by interacting with conserved carboxyl terminus of Neh2 domain. BTB, broad complex, tram-track and bric-a-brac; CTR, C-terminal region; Cul3, Cullin3; IVR, intervening region; KEAP1, kelch-like ECH-associated protein 1; sMaf, musculoaponeurotic fibrosarcoma oncogene; Neh, NRF2-ECH homologous structure; NRF2, nuclear factor erythroid-2-related factor-2; NTR, N-terminal region; RXRα,.

**Figure 2 antioxidants-09-00193-f002:**
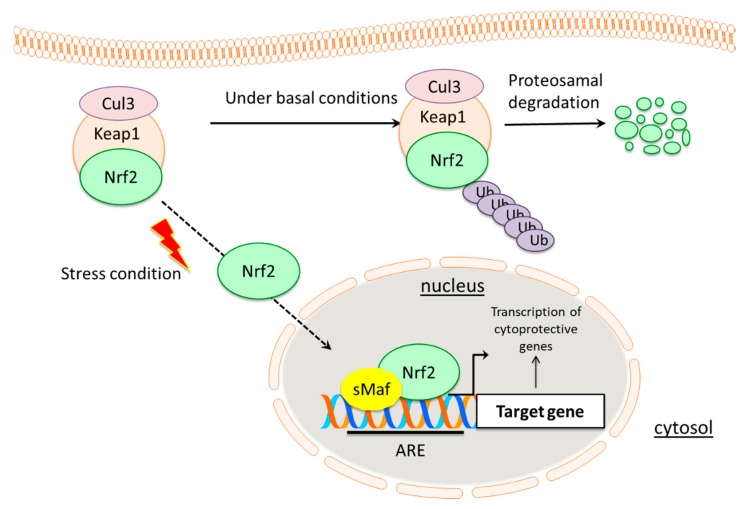
NRF2 regulation by KEAP1. Under basal conditions, NRF2 is constantly ubiquitinated through KEAP1 and degraded in the proteasome in cytosol. Under stress conditions, KEAP1-NRF2 interaction is stopped and free NRF2 translocates into nucleus. Then, NRF2 forms heterodimers with sMaf and binds to ARE sites within regulatory sites of antioxidant and detoxification genes. ARE, antioxidant response element; KEAP1, Kelch-like ECH-associated protein1; NRF2, nuclear erythroid-2 like factor-2; Retinoic X receptor alpha sMafs, small musculoaponeurotic fibrosarcoma oncogene family.

**Figure 3 antioxidants-09-00193-f003:**
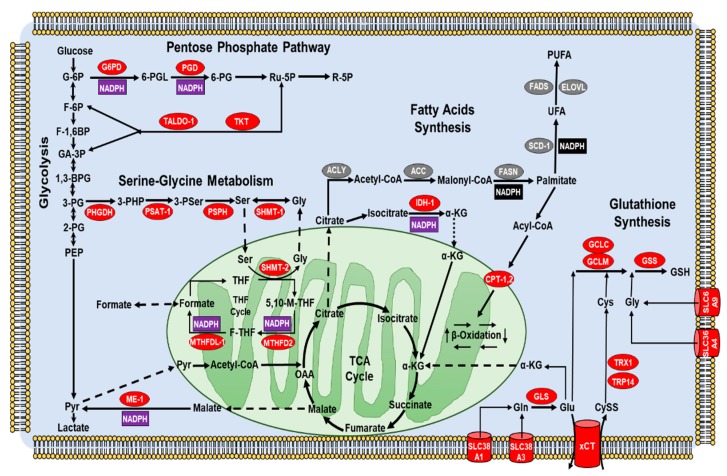
NRF2 rewires cancer cells metabolism to support the redox homeostasis. The enzymes marked in red are positively regulated while those in green are negatively modulated by NRF2. NADPH (reduced Nicotinamide adenine dinucleotide phosphate) production is indicated in violet while NADPH consumption in black. The abbreviations are: ACC1, Acetyl-CoA Carboxylase 1; ACL, ATP-Citrate Lyase; CPT, Carnitine PalmitoylTransferase; ELOVL, fatty acid Elongase; FADS, Fatty Acid Desaturase; FASN, Fatty Acid Synthase; G6PD, Glucose-6-Phosphate Dehydrogenase; GCLC, Glutamate-Cysteine Ligase, Catalytic subunit; GCLM, Glutamate-Cysteine Ligase, Modifier subunit; GLS, Glutaminase; GS, Glutathione Synthetase; IDH1, Isocitrate Dehydrogenase 1; ME1, Malic Enzyme 1; MTHFDL, MethyleneTetraHydroFolate Dehydrogenase 2; PGD, 6-phosphogluconate dehydrogenase; PHGDH, phosphoglycerate dehydrogenase; PPAT, phosphoribosyl pyrophosphate amidotransferase; PSAT1, phosphoserine aminotransferase; PSPH, phosphoserine phosphatase; SCD1, stearoyl CoA desaturase; SHMT 1-2, serine hydroxymethyltransferase 1 and 2; TALDO, transaldolase; TKT, transketolase; TXN, thioredoxin; UCP3, uncoupling protein 3; xCT, glutamate/cystine antiporter. G6P, glucose-6-phosphate; F6P, fructose-6-phosphate; F1,6BP, fructose-1,6-bisphosphate; GA3P, glyceraldehyde-3-phosphate; 3PG, 3-phosphoglycerate; PEP, phosphoenol pyruvate. PPP: 6PGL, 6-phosphoglucono-d-lactone; 6PG, 6-phosphogluconate. PRPP, 5-phospho-D-ribosyl-1 pyrophosphate; IMP, inosine monophosphate. Ser/Gly synthesis: 3PHP, 3 phosphohydroxypyruvate; 3PSer, 3-phosphoserine; THF, tetrahydrofolate; MTHF, methylenetetrahydrofolate; 5,10-FTHF, 5,10-methenyl-tetrahydrofolate. b-Oxidation: Acyl-CoA, acyl-coenzyme A; Ac-CoA, acetyl-coenzyme A. FA, fatty acid. GSH, glutathione, reduced.

**Table 1 antioxidants-09-00193-t001:** Selected list of NRF2 target genes.

Gene	Coded Protein	Functional Category	Biological Role	Ref.
*GCLC*	Glutamate-Cysteine Ligase Catalytic Subunit	GSH Synthesis & Regeneration	GSH Synthesis	[51]
*GCLM*	Glutamate-Cysteine Ligase Modulatory Subunit	GSH Synthesis & Regeneration	GSH Synthesis	[51]
*GSR1*	Glutathione Reductase 1	GSH Synthesis & Regeneration	GSH Regeneration	[51]
*SLC7A11*	xCT, Light Subunit of Xc-Antiporter	GSH Synthesis & Regeneration	Cystine Uptake	[51]
*PHGDH*	Phosphoglycerate Dehydrogenase	GSH Synthesis & Regeneration	Serine/Glycine Synthesis	[21]
*PSAT1*	Phosphoserine Aminotransferase 1	GSH Synthesis & Regeneration	Serine/Glycine Synthesis	[21]
*PSPH*	Phosphoserine Phosphatase	GSH Synthesis & Regeneration	Serine/Glycine Synthesis	[21]
*SHMT 1,2*	Serine Hydroxymethyltransferase 1,2	GSH Synthesis & Regeneration	Serine/Glycine Synthesis	[21]
*GPX1,2,4*	Glutathione Peroxidase 1,2,4	ROS & Phase-II Xenobiotic Detoxification	ROS Scavenging	[21,51]
*PRDX1,6*	Peroxiredoxin 1,6	ROS & Phase-II Xenobiotic Detoxification	ROS Scavenging	[52]
*TXN1*	Thioredoxin 1	Thioredoxin-linked Antioxidant Role	Reduction of Sulfenylated-Proteins	[51]
*TXNRD1*	Thioredoxin Reductase-1	Thioredoxin-linked Antioxidant Role	Reduction of Thioredoxin	[51]
*NQO1*	NAD(P)H dehydrogenase Quinone 1	ROS & Phase-I Xenobiotic Detoxification	Reduction of quinones	[51,53]
*AKR1B1*	Aldo-Keto Reductase Family 1 Member B1	Phase-I Xenobiotic Detoxification	Reduction of aldehydes and ketones	[52]
*AKR1B10*	Aldo-Keto Reductase Family 1 Member B10	Phase-I Xenobiotic Detoxification	Reduction of aldehydes and ketones	[52]
*AKR1C1*	Aldo-Keto Reductase Family 1 Member C1	Phase-I Xenobiotic Detoxification	Reduction of aldehydes and ketones	[52]
*ALDH1A1*	Aldehyde Dehydrogenase 1 Family Member A1	Phase-I Xenobiotic Detoxification	Conversion of aldehydes to carboxylic acids	[54,55]
*ALDH3A1*	Aldehyde Dehydrogenase 3 Family Member A1	Phase-I Xenobiotic Detoxification	Conversion of aldehydes to carboxylic acids	[52]
*GSTA 1,2,3,5*	Glutathione-S Transferase A1,2,3,5	ROS & Phase-II Xenobiotic Detoxification	Conjugation of Glutathione to electrophiles	[51]
*GSTM 1,2,3*	Glutathione-S Transferase M1,2,3	ROS & Phase-II Xenobiotic Detoxification	Conjugation of Glutathione to electrophiles	[51]
*UGT1A1,5*	UDP Glucuronosyltransferase 1 A1,5	Phase-II Xenobiotic Detoxification	Conjugation of Glucuronic acid to electrophiles	[52]
*ABCC1*	ATP Binding Cassette Subfamily C Member 1/Multidrug resistance associated protein 1 (MRP1)	Phase-III Xenobiotic Detoxification	Transmembrane translocation of xenobiotics	[56]
*ABCG2*	ATP Binding Cassette Subfamily G Member 2	Phase-III Xenobiotic Detoxification	Transmembrane xenobiotic transporter	[57]
*ABCB6*	ATP Binding Cassette Subfamily B Member 6	Phase-III Xenobiotic Detoxification/Heme Synthesis	Transmembrane transport of xenobiotics and porphyrins	[52]
*ABCC2*	ATP Binding Cassette Subfamily C Member 2	Phase-III Xenobiotic Detoxification	Transmembrane transport of xenobiotics	[52]
*SRXN1*	Sulfiredoxin-1	Thioredoxin-linked Antioxidant Role	Reduction of Sulfinylated-Peroxiredoxins	[51]
*G6PD*	Glucose-6-Phosphate Dehydrogenase	NADPH Generation	Pentose Phosphate Pathway/Glucose to Glucose 6-Phosphate Conversion	[53]
*PGD*	6-Phosphogluconate Dehydrogenase	NADPH Generation	Pentose Phosphate Pathway/6-Phosphogluconate to Ribulose 5-Phosphate Conversion	[53]
*ME1*	Malic Enzyme 1	NADPH Generation	Malate to Pyruvate Conversion	[53]
*IDH1*	Isocitrate Dehydrogenase 1	NADPH Generation	Isocitrate to α-Ketoglutarate Conversion/TCA Cycle	[53]
*TKT*	Transketolase	NADPH Generation	Pentose Phosphate Pathway/Conversion of Xylulose 5-Phosphate and Ribose 5-Phosphate into Glyceraldehyde 3-Phosphate and Sedoheptulose 7-Phosphate	[53]
*TALDO1*	TransAldolase 1	NADPH Generation	Pentose Phosphate Pathway/Conversion of Glyceraldehyde 3-Phosphate and Sedoheptulose 7-Phosphate into Erythrose 4-Phosphate and Fructose 4-Phosphate	[53]
*MTHFD2*	Methylenetetrahydrofolate Dehydrogenase 2	NADPH Generation	Serine/Glycine Metabolism	[53]
*MTHFDL1*	Methylenetetrahydrofolate Dehydrogenase 1-like	NADPH Generation	Mitochondrial Tetrahydrofolate Synthesis	[58]
*HMOX1*	Heme Oxygenase 1	Heme & Iron Metabolism	Heme to Biliverdin Conversion	[51]
*FTL*	Ferritin Light Chain	Heme & Iron Metabolism	Iron Storage	[51]

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
