# Peer review of "Potential Applications of NRF2 Modulators in Cancer Therapy"

_antioxidants, 2020, doi:10.3390/antiox9030193_

Round 1

Reviewer 1 Report

This review is focused on The NRF2/KEAP1 regulatory pathway plays an essential role in protecting cells and tissues from oxidative, electrophilic and xenobiotic stress. Specially, Nrf2/Keap1 system exerts for potential therapeutic strategies based on the use of context-specific modulation of NRF2. This is interesting scientific review. I think this one is an acceptable.

Author Response

We appreciate the positive comments of the reviewer.

Reviewer 2 Report

I suggest the authors to change the term antioxidant genes by genes encoding proteins involved in the antioxidant cycle, or a similar statement as the genes are not antioxidant entities per se.

It would be interesting to the readers to define the term hyperactivation (line 66) as it seems a circumstance correlated with the cancer promotion activity of the NRF2.

A reference should be included in lines 141-142 regarding the 500 target genes nature, or directly go to the Table 1, that should include the literature references in a single column where the NRF2 promoting activity towards those genes has been shown.

The reference in line 191 should be included as a number according to the journal's instructions.

There should be clarified whether the CDDO is a natural product or a synthesis-yield from oleanolic acid (line 207).

Should the references in Table 2 denoted as numbers?

The authors should clarify whether the NRF2 activation promotes the KRAS, BRAF and MYC oncogenes or viceversa (lines 410-412).

Author Response

Reviewer 2: I suggest the authors to change the term antioxidant genes by genes encoding proteins involved in the antioxidant cycle, or a similar statement as the genes are not antioxidant entities per se.

Authors: We agree with the Reviewer that it is better to make a distinction between genes and proteins. In the text the term “antioxidant genes” has been now substituted with “genes encoding for antioxidant enzymes” or with “genes and their encoded antioxidant proteins”.

Reviewer 2: It would be interesting to the readers to define the term hyperactivation (line 66) as it seems a circumstance correlated with the cancer promotion activity of the NRF2.

Authors: We appreciate the reviewer’s comment. Since the extent of NRF2 activation can differ from cancer to cancer, it is quite difficult to define in a quantitative way the degree of NRF2 induction underlined by the term “hyperactivation”. Several pathways involving oncogenic signaling have been investigated as the cause of Nrf2 hyperactivation. Cancer cells can hijack the NRF2 signaling pathway for their survival through mechanisms and pathways that can promote constitutive activation of NRF2 signaling, such as somatic mutations of Nrf2-Keap1 system, aberrant epigenetic alterations and electrophilic attack by oncometabolites. It has been proposed that the MAP kinases ERK and JNK (S. Papaiahgari et al. 2004; X. Yuan et al. 2006;  R. Yu et al. 2000; S.B. Cullinan et al. 2003), PI3K/Akt (K.C. Kim et al. 2010) and PKC (H.C. Huang et al. 2002) can hyperactivate the Nrf2-mediated antioxidant response. By the term “hyperactivation” it is meant the increased transcriptional activity of Nrf2 that can be modulated by these pathways. As a consequence, NRF2 hyperactivation may promote cell proliferation, confer therapy-resistance to cancer cells and lead to metabolic reprogramming, largely mediating typical hallmarks of malignant cells. In conclusion,  we think that it is more appropriate to substitute the term ”hyperactivation”, with a more generic “increased/enhanced NRF2 transcriptional activity”, which is a condition shared by the vast majority of malignant tumors.

Reviewer2: A reference should be included in lines 141-142 regarding the 500 target genes nature, or directly go to the Table 1, that should include the literature references in a single column where the NRF2 promoting activity towards those genes has been shown.

Authors: The references confirming the activation of more than or nearly 500 target genes have been added now in the main text. Also, a new column with the literature references reporting the activation of the indicated genes, has been added in Table 1.

Reviewer2: The reference in line 191 should be included as a number according to the journal's instructions.

Authors: The indicated reference has been now formatted.

Reviewer2: There should be clarified whether the CDDO is a natural product or a synthesis-yield from oleanolic acid (line 207).

Authors: We agree with the reviewer that this point needs to be clarified. The CDDO is a synthetic derivative of the natural triterpenoid oleanolic acid (the same is true for CDDO-Me). This statement has been now implemented in the line 207.

Reviewer2: Should the references in Table 2 denoted as numbers?

Authors: We preferred to keep the references in the current form, if this is not a problem

Reviewer2: The authors should clarify whether the NRF2 activation promotes the KRAS, BRAF and MYC oncogenes or viceversa (lines 410-412).

Authors: We agree with the reviewer that this period should be rephrased to better elucidate the functional relationship between NRF2 and the KRAS, BRAF, MYC oncogenes. We have now clarified that these oncogenes can stimulate the transcription of NRF2 (but not viceversa, to our best knowledge), in agreement with the study of De Nicola GM et al. 2011, wherein it is shown that K-RasG12D and B-RafV619E induce the transcription of Nrf2 via Jun and Myc.

Reviewer 3 Report

Authors provided a roboust review of research on the Nrf2-Keap1 pathway with focus on cancer environment. This red-ox sensitive pathway evolved to protect a cell against oxidative insults or toxic, unknown substances (xenobiotics) and to promote survival. For these reasons, it is a perfect pathway to be exploited by cancer cells. Indeed, tumors overuse Nrf2 signalling to sustain their survival and make themselves independent from death promoting signals. In many types of tumors, especially lung cancer or melanoma, Nrf2 is overexpressed, often thanks to Keap1 mutations, and drives cancer cell proliferation and metabolism. That's way we deal with two contradictory approaches to this two-faced pathway - Nrf2 activation for cancer preventive purposes (chemoprevention) and Nrf2 inhibition in already established tumors or pre-cancer states. However simple it sounds, it is very challenging to apply clinically. Especilly due to the fact that so far majority of clinical trials have tested potential Nrf2 activators (mainly natural compounds like curcumin, EGCG, PolyE) from the perspective of alleviation side effects of standard radio- or chemotherapy and Nrf2 inhibitors for cancer treatment have not been tested clinically so far.

This issue was presented by the authors in a very detailed and careful way, with the great focus on compounds - natural or synthetic - which modulate the Nrf2/Keap1 pathway both ways.

There are maybe two things missing for me in this article, but of minor significance:

  1. The anti-inflammatory action of DMF, potential Nrf2 activator, was shown to be Nrf2 independent by Ulf Schulze-Topphoff et al. (2016) . Maybe it is worth mentioning, also in this review, that Nrf2 pathway might not be the only pathway resposible for the observed effects of DMF.
  2.  In the second part, where authors present potential Nrf2 inhibitors for cancer therapy, i did not find information on ML385, a molecule shown by Singh et al. (2017) to successfully inhibit Nrf2 activity in cancer cells overexpressing Nrf2 like A549 (doi: 10.1021/acschembio.6b00651)

Apart from these minor concerns i do believe that presented manuscript is a source o valuable and up-to-date information on Nrf2 signalling in healthy and cancer state. It is also a compendium of molecules, natural and synthetic, which alter this signalling - very beneficial in studies addressing the role of oxidative stress modulators in cancer or anticancer potential of natural compounds.

Author Response

Reviewer3: There are maybe two things missing for me in this article, but of minor significance:

  1. The anti-inflammatory action of DMF, potential Nrf2 activator, was shown to be Nrf2 independent by Ulf Schulze-Topphoff et al. (2016) https://doi.org/10.1073/pnas.1603907113. Maybe it is worth mentioning, also in this review, that Nrf2 pathway might not be the only pathway resposible for the observed effects of DMF.
  2.  In the second part, where authors present potential Nrf2 inhibitors for cancer therapy, i did not find information on ML385, a molecule shown by Singh et al. (2017) to successfully inhibit Nrf2 activity in cancer cells overexpressing Nrf2 like A549 (doi: 10.1021/acschembio.6b00651)

Authors: We appreciate the positive comments of the reviewer and his/her suggestions.

1. We also agree that it is important to indicate that the compound DMF might promote both NRF2-dependent as well NRF2 independent mechanisms. For this reason, we have now added this statement and the suggested article in the specific subsection (see lines 394-396).

2. Also we recognize that the compound ML385 has revealed a significant therapeutic potential in the treatment of lung cancers. For this reason we have now included the ML385 in the list of NRF2 inhibitors (see lines 802-812).